# Reversible epigenetic alterations regulate class I HLA loss in prostate cancer

Tamara S. Rodems [1,9], Erika Heninger[1,2,9], Charlotte N. Stahlfeld[1], Cole S. Gilsdorf[1], Kristin N. Carlson[1], Madison R. Kircher[1], Anupama Singh[1,2], Timothy E. G. Krueger[1], David J. Beebe [1,3,4], David F. Jarrard[1,5], Douglas G. McNeel [1], Michael C. Haffner[6,7,8] & Joshua M. Lang [1,2✉]

Downregulation of HLA class I (HLA-I) impairs immune recognition and surveillance in prostate cancer and may underlie the ineffectiveness of checkpoint blockade. However, the molecular mechanisms regulating HLA-I loss in prostate cancer have not been fully explored. Here, we conducted a comprehensive analysis of HLA-I genomic, epigenomic and gene expression alterations in primary and metastatic human prostate cancer. Loss of HLA-I gene expression was associated with repressive chromatin states including DNA methylation, histone H3 tri-methylation at lysine 27, and reduced chromatin accessibility. Pharmacological DNA methyltransferase (DNMT) and histone deacetylase (HDAC) inhibition decreased DNA methylation and increased H3 lysine 27 acetylation and resulted in re-expression of HLA-I on the surface of tumor cells. Re-expression of HLA-I on LNCaP cells by DNMT and HDAC inhibition increased activation of co-cultured prostate specific membrane antigen (PSMA)$_{27-38}$-specific CD8$^+$ T-cells. HLA-I expression is epigenetically regulated by functionally reversible DNA methylation and chromatin modifications in human prostate cancer. Methylated HLA-I was detected in HLA-I$^{low}$ circulating tumor cells (CTCs), which may serve as a minimally invasive biomarker for identifying patients who would benefit from epigenetic targeted therapies.

[1] University of Wisconsin Carbone Cancer Center, University of Wisconsin, Madison, 1111 Highland Ave., Madison, WI 53705, USA. [2] Department of Medicine, University of Wisconsin, Madison, 1111 Highland Ave., Madison, WI 53705, USA. [3] Department of Biomedical Engineering, University of Wisconsin, Madison, 1111 Highland Ave., Madison, WI 53705, USA. [4] Department of Pathology, University of Wisconsin, Madison, 3170 UW Medical Foundation Centennial Building, 1685 Highland Ave., Madison, WI 53705, USA. [5] Department of Urology, University of Wisconsin, Madison, 1111 Highland Ave., Madison, WI 53705, USA. [6] Divisions of Human Biology and Clinical Research, Fred Hutchinson Cancer Research Center, 1100 Fairview Ave, N., Seattle, WA 98109, USA. [7] Department of Pathology, University of Washington, 1959 NE Pacific St., Seattle, WA 98195, USA. [8] Department of Pathology, Johns Hopkins School of Medicine, 600N Wolfe St., Baltimore, MD 21287, USA. [9] These authors contributed equally: Tamara S. Rodems, Erika Heninger. ✉email: jmlang@medicine.wisc.edu

Approximately 30,000 men die of metastatic prostate cancer per year in the United States and the incidence of men presenting with metastatic disease is rising[1,2]. There is a critical need to identify the molecular drivers that contribute to prostate cancer growth and metastasis. Immune evasion is one of the hallmarks of cancer pathogenesis and can be therapeutically targeted by immunotherapies that augment T-cell recognition and lysis of tumor cells[3–5]. However, display of a functional major histocompatibility complex class I (MHC-I) is required for recognition of tumor cells by cytotoxic T lymphocytes. Lack of MHC-I display at the cell surface reduces tumor immunogenicity and drives resistance to immune checkpoint inhibitors[6–9]. MHC-I is a multimeric protein composed of a class I human leukocyte antigen (HLA-I) protein (A, B, or C), beta-2-microglobulin (B2M), and a peptide derived from an intracellular protein[10]. Downregulation of the MHC-I components has been proposed as a mechanism of immune evasion in numerous cancer types, including prostate cancer[6,11,12]. Downregulation of HLA-I was observed in ~70% of primary prostate tumors with complete loss in up to 34% of primary tumors and 80% of metastatic lesions[13,14]. The molecular alterations that lead to HLA-I downregulation in prostate cancer remain largely unknown.

Recent findings have pointed to epigenetic mechanisms as drivers of prostate cancer progression[15,16]. Globally, DNA methylation at specific gene promoters increases through progression, often resulting in silencing of genes involved in cell cycle progression and carcinogenesis[17]. Altered levels of certain histone modifications have also been reported in prostate cancer and have been reported to be useful as predictive biomarkers[18]. Overexpression of epigenetic modifying proteins, including the de novo methyltransferases, DNMT3A and DNMT3B, and class I histone deacetylases, HDAC1, HDAC2, and HDAC3, has been implicated in altering epigenetic programs and contributing to metastasis in prostate cancer[19,20]. Studies in esophageal squamous cell carcinoma and gastric cancer found that DNA methylation contributed to HLA-I downregulation, however, epigenetic regulation of HLA-I has not yet been explored in prostate cancer[21,22]. Inhibition of DNMT and HDAC proteins has been proposed as a therapeutic strategy in prostate cancer, though minimal clinical success has been observed in solid tumors[23]. Despite this, there is promising evidence for the usefulness of epigenetic therapies in combination with immunotherapy[24]. There remains a need to better understand the interplay between epigenetic and immune functions in cancer cells and for accessible biomarkers for both predicting treatment benefit and monitoring treatment response[24]. We have previously demonstrated that modulating the activity of DNMT and HDAC proteins in prostate cancer cell lines and ex vivo human prostate tissue induced the expression of cancer testis antigens (CTAs), which have been proposed as targets for tumor vaccine therapies[25]. Taking advantage of the ability of epigenetic mechanisms to modulate aspects of the immune response, such as HLA-I expression, may improve the efficacy of certain immunotherapies.

In this report, we demonstrate that transcriptional downregulation of HLA-I is coordinated by epigenetic silencing mechanisms, which can be reversed to functionally re-express HLA-I and restore MHC-I-dependent T-cell activation. In addition, we show that methylation in the HLA-I genes can be detected in prostate cancer circulating tumor cells (CTCs) with low MHC-I protein expression, which is a potential predictive biomarker to identify patients who may benefit from epigenetic therapies and monitor treatment response.

## Results

**HLA-I gene expression is downregulated in a subset of prostate cancers.** Previous studies have shown widespread HLA-I downregulation at the protein level in prostate cancer tissues when compared to normal tissue[12–14,26]. To confirm that this phenomenon occurs at the transcriptional level, we analyzed changes in HLA-I gene expression in prostate cancer compared to normal tissue in two data sets that included normal tissue in their gene expression analysis: the TCGA PanCancer Prostate Adenocarcinoma (PRAD) data set, which contains only primary tumor samples, and a data set from Taylor et al., which contained both primary and metastatic tumor samples[27–29]. Gene expression in the PRAD data set was generated with RNA-seq and gene expression in the Taylor data set was generated with microarrays. Data were converted into z-scores comparing tumor samples to the normal samples from the respective study in order to make comparisons between the two different experimental systems, however, appropriate precaution should be used for any primary:primary comparisons due to the different methods of data acquisition.

We examined both the mean z-score in the population as well as the number and percentage of samples that were significantly up- and downregulated based on a confidence level of 95% (Fig. 1a; Supplementary Data S1). In the two primary data sets, only HLA-A showed noticeable negative shift in mean z-score, indicating a population level downregulation of gene expression compared to normal samples. However, for all three genes, there was a subset of the population ranging from 5 to 14% with significantly reduced gene expression in the primary tumors (Supplementary Data S1). HLA-A, HLA-B, and HLA-C were downregulated in 58–63% of the metastatic samples and had significantly negative shifts in mean z-score at the population level. In addition, expression of each HLA-I gene was highly positively correlated with the other HLA-I genes, suggesting no compensation from the other genes takes place if one is lost (Fig. 1b).

Significant downregulation of HLA-I was also associated with decreased time to biochemical recurrence (two occurrences of PSA ≥ 0.2 ng/mL) after radical prostatectomy when compared to all other patients or patients with significantly upregulated HLA-I expression (Fig. 1c, d). These data show that HLA-I may be transcriptionally downregulated in a subset of primary prostate cancers and the majority of metastatic prostate cancers and loss of HLA-I gene expression may be a risk factor for biochemical recurrence.

**Aberrantly expressed DNMT and class I HDAC genes are correlated to HLA-I in prostate cancer.** Analysis of copy number alterations and mutations in the HLA-I genes revealed that the majority of HLA-I gene expression changes in prostate cancer are unlikely to be driven by genomic alterations due to low alteration frequency and lack of correlation to gene expression changes (Fig. S1). Previous studies have identified epigenetic mechanisms, including DNA methylation and histone modifications, as driving forces in prostate cancer biology[15,16]. DNMTs and class I HDACs have been widely studied as potential therapeutic targets as well as mechanistic drivers for regulation of gene expression[19,20]. We hypothesized that the DNMTs and class I HDACs could promote immune evasion by downregulating HLA-I. We analyzed the expression of DNMT1, DNMT3A, DNMT3B, and the class I HDACs HDAC1, HDAC2, HDAC3, and HDAC8 in the PRAD and Taylor data sets and their correlation to HLA-I expression in prostate cancer. Supplementary Data S1 summarizes the number and percentage of patients in each study who showed up- and downregulation of each of these genes. The de novo DNMTs and class I HDAC genes tended to be upregulated more often than downregulated with the exception of HDAC8, which was upregulated in 25% of the samples in the PRAD data set, but only in

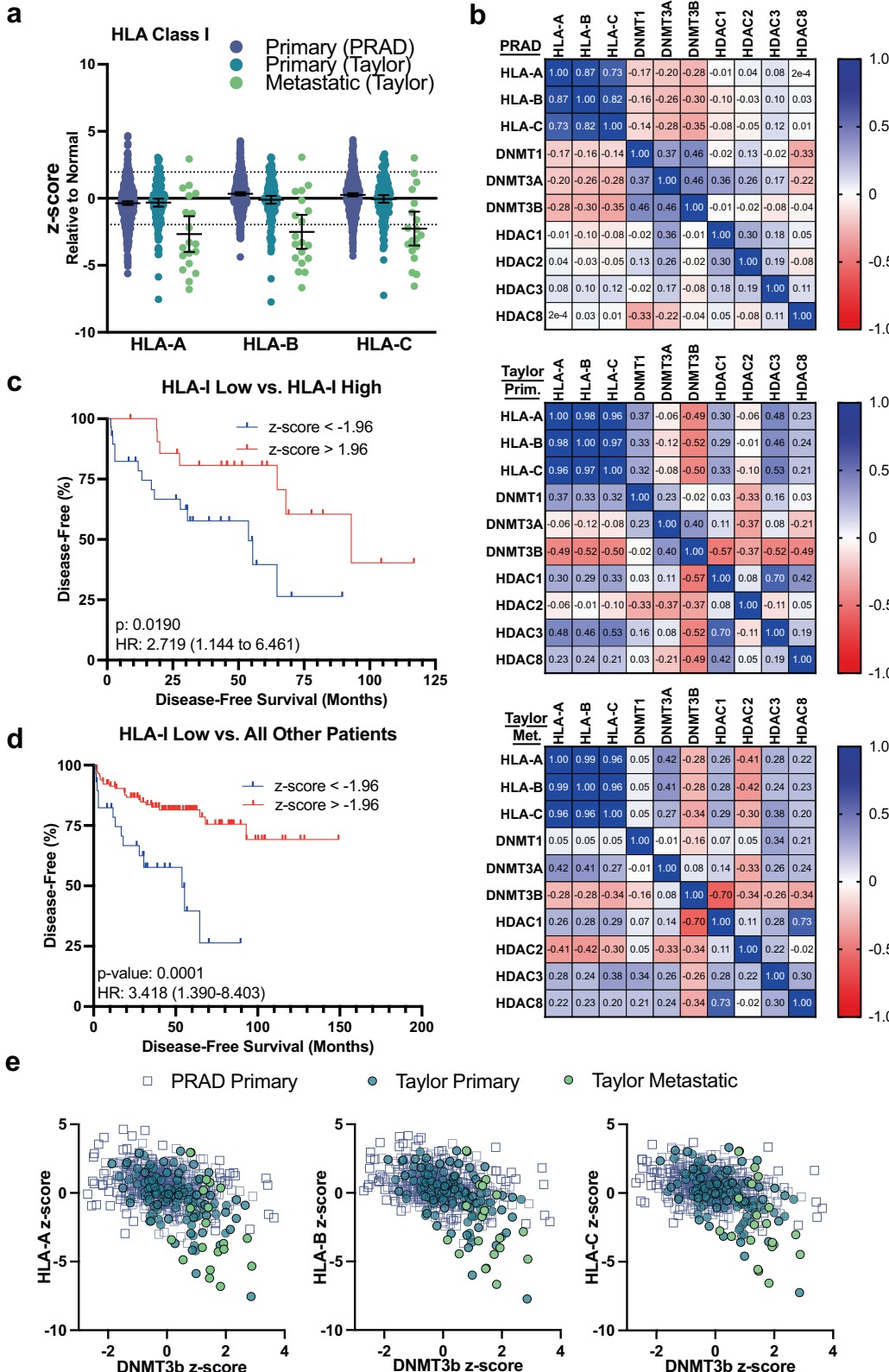

**Fig. 1 Genomic alterations and gene expression of HLA-I in primary and metastatic prostate adenocarcinoma. a** HLA-A, HLA-B, and HLA-C gene expression in the TCGA-PRAD data set (PRAD, $n = 497$, blue circle), and Taylor data set (Primary $n = 131$, teal circle; Metastatic $n = 19$, green circle) represented as z-scores relative to normal samples in the respective study. Dotted lines are at $+/−1.96$. Line and error bars represent mean and 95% confidence interval. **b** Correlation matrix of gene expression of HLA-I genes and epigenetic modifying proteins in PRAD data set. **c** Kaplan–Meier curves showing disease-free survival time for patients in the Taylor data set with low expression (z-score $< −1.96$; $n = 29$) vs. high expression (z-score $>1.96$, $n = 22$) of any HLA-I gene. **d** Kaplan–Meier curves showing disease-free survival time for patients in the Taylor data set with low HLA-I expression in any HLA-I gene (z-score $< −1.96$, $n = 29$) vs. all other patients (z-score $> −1.96$, $n = 116$). **e** Correlation of expression of *DNMT3b* to HLA-I genes.

1 sample from the Taylor data set. An analysis of the mean z-score in each data set revealed that *DNMT3A*, *DNMT3B*, and *HDAC1/2/3* tended to have positively shifted z-scores indicating an increase in mean gene expression in the tumor populations (Fig. S2a).

To investigate how the expression of these various genes associates with HLA-I expression, we calculated Pearson r values and *p*-values for the correlation between HLA-I and DNMT and HDAC genes in each data set (Fig. 1b and Supplementary Data S2). Correlation patterns varied among the data sets, however, we found that *DNMT3B* was highly negatively correlated to HLA-I gene expression in all three data sets and *HDAC2* was the most negatively correlated to HLA-I gene expression in the metastatic samples (Figs. 1e and S2b). *DNMT3A* was also negatively correlated to HLA-I gene expression in the PRAD data set. *HDAC3* and *HDAC8* gene expression was positively correlated to HLA-I gene expression in the Taylor data sets, though showed weak to no correlation in the PRAD data set. Overall, the correlation of *DNMT3B* and *HDAC2* to HLA-I gene expression along with overexpression of these gene families, suggests a key role for epigenetic modification of DNA and histones in HLA-I downregulation.

**HLA-I CpG islands are methylated in prostate cancer**. We next sought to investigate the DNA methylation signatures of the HLA-I genes in primary and metastatic prostate cancer biopsies. We analyzed the level of methylation at individual probes from the Illumina 450 K methylation array within the HLA-I CpG islands in normal prostate tissue vs. prostate adenocarcinoma samples in the PRAD data set (Fig. S3a, b). Probe locations are shown in Fig. 2a. Every probe located within the *HLA-A* CpG island showed higher methylation levels in prostate tumor samples compared to normal. There were also multiple probes located within *HLA-B* and *HLA-C* with significantly higher levels of methylation compared to normal samples. Notably, the probe located within 50 bp upstream of the transcription start site for both *HLA-A* and *HLA-C* had the most significant increase in methylation in prostate tumors compared to normal samples (Fig. S3b). In order to examine overall methylation levels in the promoter and intragenic regions of the CpG islands, average methylation level at the probes within these two regions was analyzed for tumor and normal samples (Fig. 2b). The average level of methylation was significantly higher in the tumor samples for all regions except the *HLA-B* promoter. *HLA-A* had the highest overall methylation level and all three genes tended to have higher methylation in the intragenic regions compared to the promoters.

We then explored whether the patients with increased levels of methylation in HLA-I had corresponding decreases in HLA-I gene expression. Correlations between matched patient gene expression from RNA-seq data and the methylation score at each probe were calculated (Supplementary Data S3). Significant negative correlations were found in 10/10 HLA-A probes, 5/13 *HLA-B* probes, and 10/11 *HLA-C* probes in the tumor samples, and only 1/10, 3/13, and 2/11 for *HLA-A*, *HLA-B*, and *HLA-C* normal samples, respectively. While many of these correlations have reached statistical significance, the Pearson *r* values are relatively weak, which may be due to the smaller subset of patients in these cohorts that have significantly reduced HLA-I gene expression, as summarized in Supplementary Data S1. To further investigate the association between gene expression and methylation in HLA-I genes, we separated the samples into three groups: high, medium, and low expression. High and low expression were defined by having a significant z-score using a 95% confidence level relative to normal samples, with all non-

significant samples placed into the medium category. We then compared methylation levels in the stratified samples to normal samples at each probe and the average methylation across the promoter and intragenic regions (Figs. 2c and S3d, e). Methylation in *HLA-A* and *HLA-C* promoter and intragenic regions was significantly higher in the samples with low HLA-I expression compared to samples with high HLA-I expression. Samples expressing medium levels of HLA-I also tended to have lower methylation levels compared to samples expressing low levels of HLA-I. The average methylation levels in the *HLA-B* promoter and intragenic regions did not show strong differences between gene expression groups. However, when we looked at individual *HLA-B* probes, we saw similar trends of low gene expression being associated with higher methylation levels (Fig. S2f). This analysis suggests a role for DNA methylation in regulating HLA-I expression in primary prostate cancer.

We next evaluated the presence of *HLA-A* methylation in primary and metastatic tumor samples (Fig. 2d). We employed COMPARE-MS, a method that has been previously used to measure methylation in prostate cancer biopsies[30–34]. *HLA-A* methylation was either not detected or detected at very low levels in benign samples. We were able to detect *HLA-A* methylation in a small subset of primary tumors and in the majority of metastatic sites. The level of detection tended to be similar across different metastatic sites from the same patient. The increase in frequency of detection of *HLA-A* in metastatic samples suggests *HLA-A* methylation may be selected for during metastasis. Taken together with the low *HLA-A* gene expression in metastatic samples shown in Fig. 1a, these analyses suggest an important role for DNA methylation in *HLA-A* transcriptional down-regulation in patients with metastatic prostate cancer.

**Decreased chromatin accessibility is associated with HLA-I downregulation in prostate cancer**. Studies on histone modifications in prostate cancer have been conducted largely at the global level with studies typically measuring overall levels of histone modification abundance[35,36]. HLA-I specific histone modification signatures have not been explored in patients with prostate cancer. However, we can make inferences about epigenetic regulation of HLA-I in prostate cancer using ATAC-seq data from the TCGA PanCancer study. We analyzed the ATAC-seq signals in available primary prostate samples from the PRAD data set at two regions associated with the HLA promoters. We then compared the changes in chromatin accessibility at these regions to matched HLA-I gene expression. While the sample size was small, those with high HLA-I gene expression tended to have higher scores for chromatin accessibility (Fig. 2e). *HLA-A* gene expression was significantly positively correlated with ATAC-seq signal at the distal promoter and *HLA-C* gene expression was significantly positively correlated with ATAC-seq signal at the proximal promoter (Fig. 2f). All other ATAC-seq signals were also positively correlated with HLA-I gene expression, but did not reach significance, possibly due to the small sample size. This preliminary analysis suggests that HLA-I downregulation may be associated with a decrease in accessible chromatin in key regulatory regions.

**HLA-I downregulation is associated with increased DNA methylation and repressive histone modifications in prostate cancer cell lines**. We next evaluated epigenetic signatures in prostate cancer cell lines where HLA-I expression is down-regulated. HLA-I protein and gene expression were down-regulated in the prostate cancer cell lines LNCaP, 22rv1, PC3, and LAPC4 when compared to a normal prostate epithelial cell line, RWPE1 (Figs. 3a–c and S4). Methylation of HLA-I was evaluated using MBD2-based enrichment of methylated DNA followed by

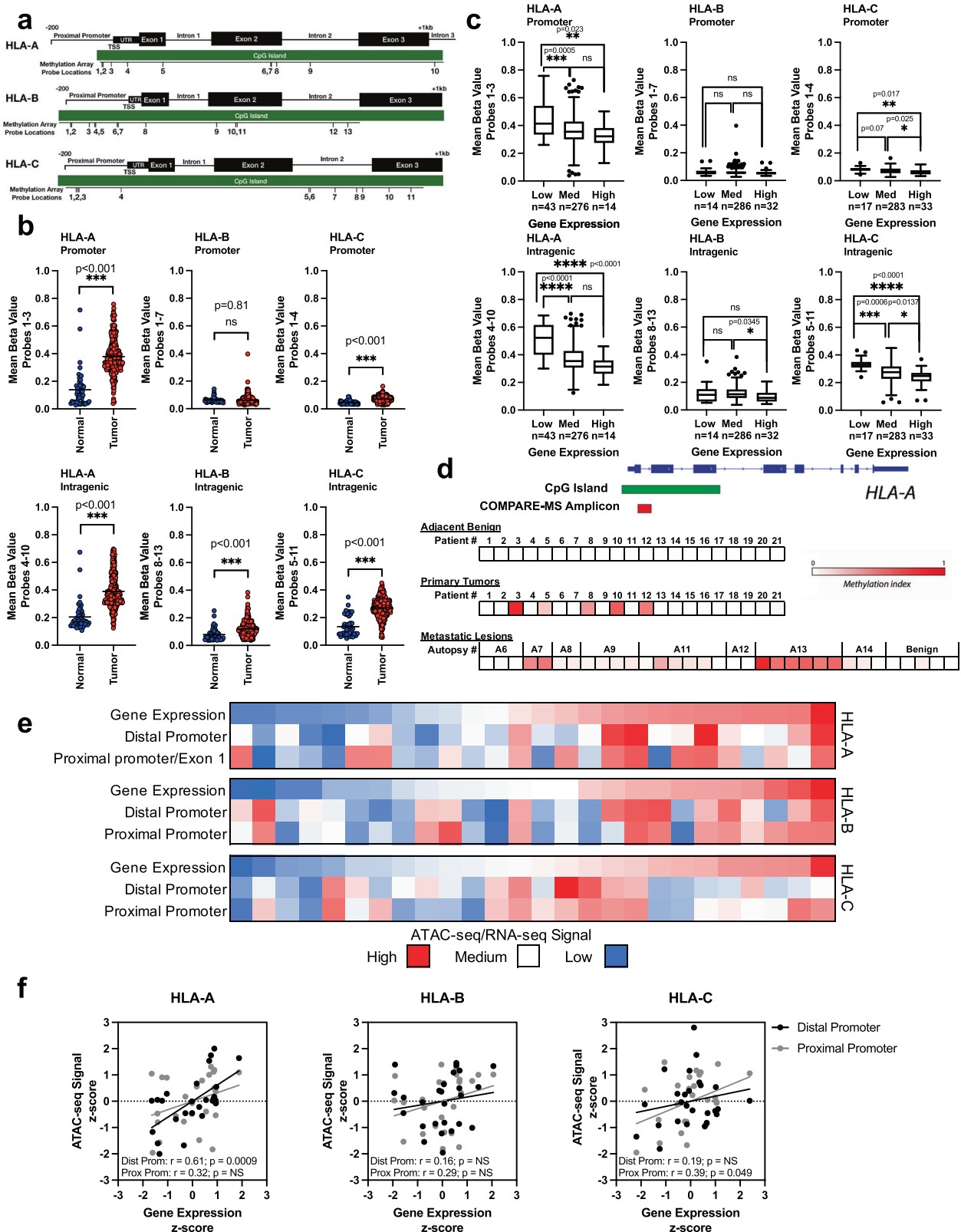

qPCR (Fig. 3d). Primers were designed to target regions of the HLA-I genes that were identified as being differentially methylated in patient samples, including the distal and proximal promoter as well as two intragenic regions (IG) near Exon1/Intron1 (IG 1) and Exon2/Intron2 (IG 2) (Fig. 3e). LAPC4 had the highest level of methylation in all three HLA-I genes. PC3 and LNCaP also had increased methylation in certain gene regions compared

to RWPE1. 22rv1 had the lowest overall methylation of the cancer cell lines. Methylation in IG2 tended to be the most enriched in cancer cell lines and overall, methylation in the cancer cell lines was increased in at least 2 of the evaluated gene regions for each gene (Fig. 3f). Overall, the methylation landscape in these four prostate cancer cell lines is comparable to the signatures found in patient samples.

**Fig. 2 DNA methylation in HLA-I genes in prostate cancer. a** Approximate locations of probes within the HLA-I CpG islands. **b** Methylation levels in the promoter or intragenic regions of the HLA-I CpG islands in normal (blue circle; $n = 49$) and primary tumor (red circle; $n = 332$ for HLA-B, $n = 333$ for HLA-A and HLA-C) samples represented by mean beta value determined by microarray analysis. **c** Methylation levels in the promoter or intragenic regions of the HLA-I CpG islands in primary tumor samples stratified by gene expression level. Low expression: z-score < −1.96, med expression 1.96 < z-score > −1.96, high expression: z-score >1.96. See figure for $n$ number for each condition. **d** Detection of HLA-A methylation in primary, adjacent benign, and metastatic lesions by COMPARE-MS. Location of amplicon is indicated. **e** ATAC-seq from PRAD data set at two promoter regions in each HLA-I gene ($n = 26$). Samples are ordered on RNA-seq signal from PRAD data set (top row on each heat map). **f** Correlation of ATAC-seq signal to gene expression in each HLA-I gene ($n = 26$). Distal promoter (black circle) proximal promoter (gray circle).

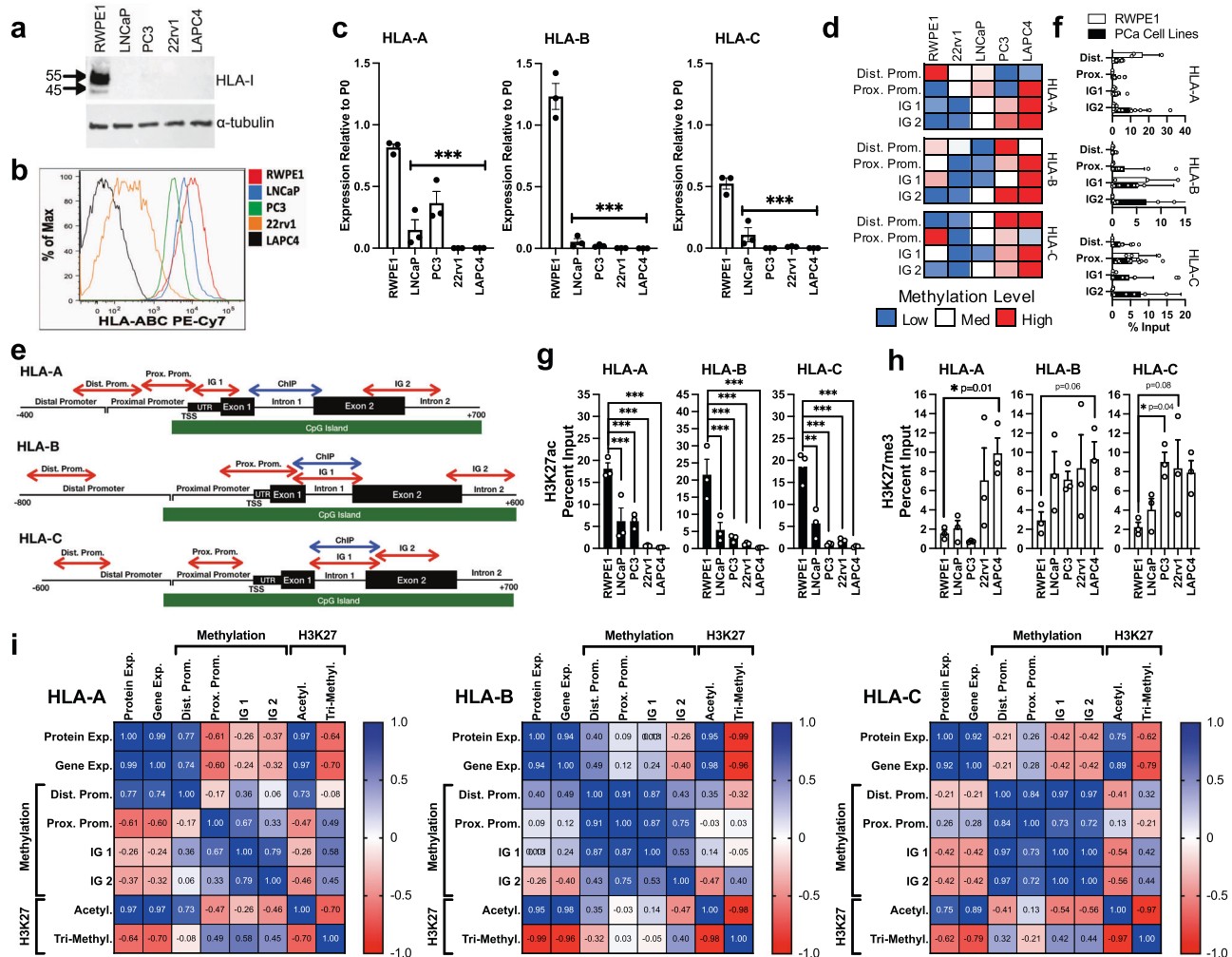

**Fig. 3 Repressive epigenetic signatures are associated with the HLA-I genes in cell lines with low HLA-I expression.** HLA-I protein and gene expression was measured in four prostate cancer cell lines (LNCaP, 22rv1, PC3, and LAPC4) and a normal prostate epithelial cell line, RWPE1. **a** Western blot analysis of HLA-I expression in whole-cell lysates with alpha-tubulin as a loading control. **b** Flow cytometry analysis of extracellular HLA-I expression. **c** Gene expression of HLA-A, HLA-B, and HLA-C measured by qPCR relative to the housekeeping gene RPLP0 (P0) ($n = 3$). **d** Heat map showing DNA methylation levels in LNCaP, 22rv1, PC3, LAPC4, and RWPE1 in four regions of the HLA-I ($n = 3$). **e** Primer locations for DNA methylation and histone modification analyses. **f** Percent input of RWPE1 methylation and averaged methylation percent input in prostate cancer (PCa) cell lines ($n = 3$). **g, h** ChIP with antibodies targeting H3K27ac ($n = 3$) (**g**) and H3K27me3 (**h**) in LNCaP, 22rv1, PC3, and LAPC4 compared to RWPE1. **i** Correlation matrix of HLA-I gene and protein expression, DNA methylation levels, and histone modifications in LNCaP, 22rv1, PC3, LAPC4, and RWPE1. Error bars represent SEM.

Since HLA-I gene expression was associated with a less accessible chromatin state (Fig. 2e), we hypothesized that repressive histone signatures may also be present in the HLA-I genes in prostate cancer cell lines. Chromatin immunoprecipitation (ChIP) was performed using antibodies targeting acetylated (H3K27ac), a marker of active transcription, and tri-methylated (H3K27me3) lysine 27 on histone H3, a marker of transcriptional repression. Primers were designed to locations near a peak in H3K27ac signature that is conserved across multiple cell lineages

determined by ChIP-seq from the ENCODE consortium[37] (Fig. 3e). This region overlaps with the proximal promoter regions surveyed by ATAC-seq in Fig. 2e. Overall, the prostate cancer cell lines showed significant decreases in H3K27ac and increases in H3K27me3 when compared to RWPE1 (Fig. 3g, h). The cell lines 22rv1 and LAPC4 had particularly low levels of lysine 27 acetylation in HLA-I. LAPC4 and 22rv1 also had the highest overall level of tri-methylation. This strong repressive signature in 22rv1 may explain why HLA-I expression is so low in

these cells, even though DNA methylation was not very high, suggesting that individual tumors may regulate HLA-I by different epigenetic mechanisms.

We generated correlation matrices for HLA-I protein expression, gene expression, methylation, and histone modifications to examine the relationships between each of these measures in the prostate cell lines (Fig. 3i). We found that protein and gene expression were highly positively correlated in each gene. Methylation within each region tended to be positively correlated with methylation in other regions and with H3K27 tri-methylation, and negatively correlated with H3K27 acetylation, indicating co-occurrence of these epigenetic signatures. The pattern of correlation between methylation signatures in each gene region and HLA-I expression was heterogeneous. We found that methylation within the IG 1 and IG 2 region of *HLA-A* and *HLA-C* was negatively correlated with corresponding gene and protein expression. *HLA-B* IG 2 methylation was also negatively correlated with HLA-B gene and protein expression. The strongest correlations were found in the histone modification group. H3K27 acetylation was strongly positively correlated with gene and protein expression and H3K27 tri-methylation was strongly negatively correlated with gene and protein expression for all three HLA-I genes. This analysis demonstrates that strong repressive epigenetic signatures are enriched at the HLA-I genes and correlated to HLA-I gene expression in prostate cancer cell lines.

**DNMT and HDAC inhibition induces HLA-I expression and reverses repressive epigenetic signatures in vitro.** To further investigate epigenetic regulation of HLA-I, we pharmacologically inhibited DNMT and HDAC activity in prostate cancer cell lines and measured HLA-I protein and gene expression in response. We treated LNCaP, 22rv1, PC3, and LAPC4 cells with DNMT inhibitors SGI-110 (SGI) and 5-aza-2-deoxycytidine (5AZA2) alone and in combination with the HDAC inhibitor LBH-589 (LBH). We found that the cancer cell lines all responded to at least one treatment condition (Fig. 4a–d). The combination treatment was the most effective to restore HLA-I expression in all cell lines tested. Overall, these results support our hypothesis that loss of HLA-I expression is regulated by epigenetic mechanisms in prostate cancer cells.

We also evaluated a panel of genes encoding antigen processing and presenting machinery (APM) proteins as well as beta-2-microglobulin (B2M) to determine whether DNMT and HDAC inhibition also increased expression of these genes. Expression of multiple APM genes, including TAP1/2 and B2M, has been demonstrated to be downregulated in prostate cancer[26]. Overall, inducibility of these genes was heterogeneous and less robust than inducibility of HLA-I (Fig. S5). Interestingly, TAP1 protein expression was induced by 5AZA2 in LNCaP and PC3 cells and LBH in 22rv1, LAPC4, and PC3, as well as the combination treatment in all cell lines tested, suggesting epigenetic induction of APM may accompany HLA-I induction in certain cell lines.

To confirm that the changes in gene and protein expression of HLA-I in response to DNMT and HDAC inhibitors are accompanied by epigenetic changes in the genes themselves, we measured DNA methylation and histone modification changes in response to SGI, 5AZA2, and LBH. HLA-I methylation was reduced across the HLA-I genes in response to SGI and 5AZA2 (Fig. 4e, f), regardless of whether gene expression was significantly induced from SGI treatment alone, suggesting DNA methylation loss may not be sufficient in all cases to re-express HLA-I. This may be due to histone modifications not changing from a repressive state even when DNA methylation has been removed. This is supported by the retention of H3K27me3 and H3K27ac

levels in 22rv1 cells when treated with SGI (Figure S6a). This can also explain why cell lines tended to show the strongest response to combination treatments. To analyze changes in histone acetylation in response to HDAC inhibition by LBH, we performed ChIP analysis to look at H3K27ac in treated cell lines. H3K27ac was significantly increased in response to LBH in 22rv1 cells (Fig. 4g) and a similar trend was observed in LNCaP and LAPC4 cells, though the results were not statistically significant.

Enrichment of RNA polymerase II (RPB1) at the HLA-I gene promoters was increased in LNCaP cells in response to SGI treatment, showing that the increase in gene expression is associated with increased transcriptional activity (Fig. 4h). RBP1 enrichment at the HLA-I promoters was also increased in 22rv1 cells in response to LBH (Fig. 4i). However, RPB1 enrichment was not increased in LNCaP cells in response to LBH treatment (Fig. S6b). This difference in LBH-induced RPB1 binding between LBH-treated 22rv1 and LNCaP cells is corroborated by the differences in inducibility of HLA-I gene expression in response to LBH in these two cell lines. While responses to DNMT and HDAC inhibition were varied in prostate cancer cell lines, the changes to the epigenetic landscape that accompany gene and protein induction suggest modulation of epigenetic proteins in prostate cancer may be useful to re-express epigenetically silenced HLA-I in patients.

**DNMT and HDAC inhibition induces HLA-I gene expression ex vivo.** The in vitro data suggest that DNMT and HDAC inhibition can alter the expression of HLA-I. We next wanted to test whether this holds true in a more physiologically relevant system. Primary prostate tumor tissue was acquired from radical prostatectomy specimens and grown in an ex vivo culture. Expression of the HLA-I genes and three prostate-specific genes, *AR*, *KLK3* (PSA), and *ACP3* (PAP), was measured to establish baseline HLA-I levels and confirm the presence of prostate cells in the culture (Fig. 5a). Tissue in ex vivo culture was treated with DMSO, 5AZA2, or LBH either alone or in combination and induction of HLA-I expression was measured (Fig. 5b). Similar to our in vitro cell culture studies, we found considerable heterogeneity regarding both the magnitude and pattern of HLA-I gene induction in response to either drug among these patient samples. 3 out of 5 patients showed robust induction in response to at least one of the treatment regimens. Among the responders, 2 out of 3 responded to 5AZA2 and showed no response to LBH alone. 1 out of 3 responders showed strong induction to LBH and less response to 5AZA2. In 2 out of 5 patient samples we saw negligible response to 5AZA2 or LBH treatments. Induction of APM genes and B2M was also variable, but tended to correlate with HLA-I induction (Fig. S7). Overall, these ex vivo culture studies corroborate the cell line data and indicate a biologically relevant role for DNMT and HDAC inhibition in re-expression of HLA-I in prostate cancer. Moreover, these data again highlight the variability in response to inhibition of these two classes of epigenetic regulators and underlines the multiple epigenetic mechanisms that are at play.

**HLA-I upregulation on tumor cells by DNMT and HDAC inhibition enhances activation of PSMA-specific T-cells.** We next sought to assess the functional relevance of HLA-I upregulation by DNMT and HDAC inhibition. To do this, we measured the CD8[+] T-cell response to LNCaP cells treated with 5AZA2, SGI, and LBH alone and in combination. PSMA$_{27-38}$-specific CD8[+] T-cells were raised by peptide vaccination in HHD mice expressing humanized HLA-A*02. HLA-A*02 expressing LNCaP cells were pretreated with 5AZA2, SGI, or LBH or a combination

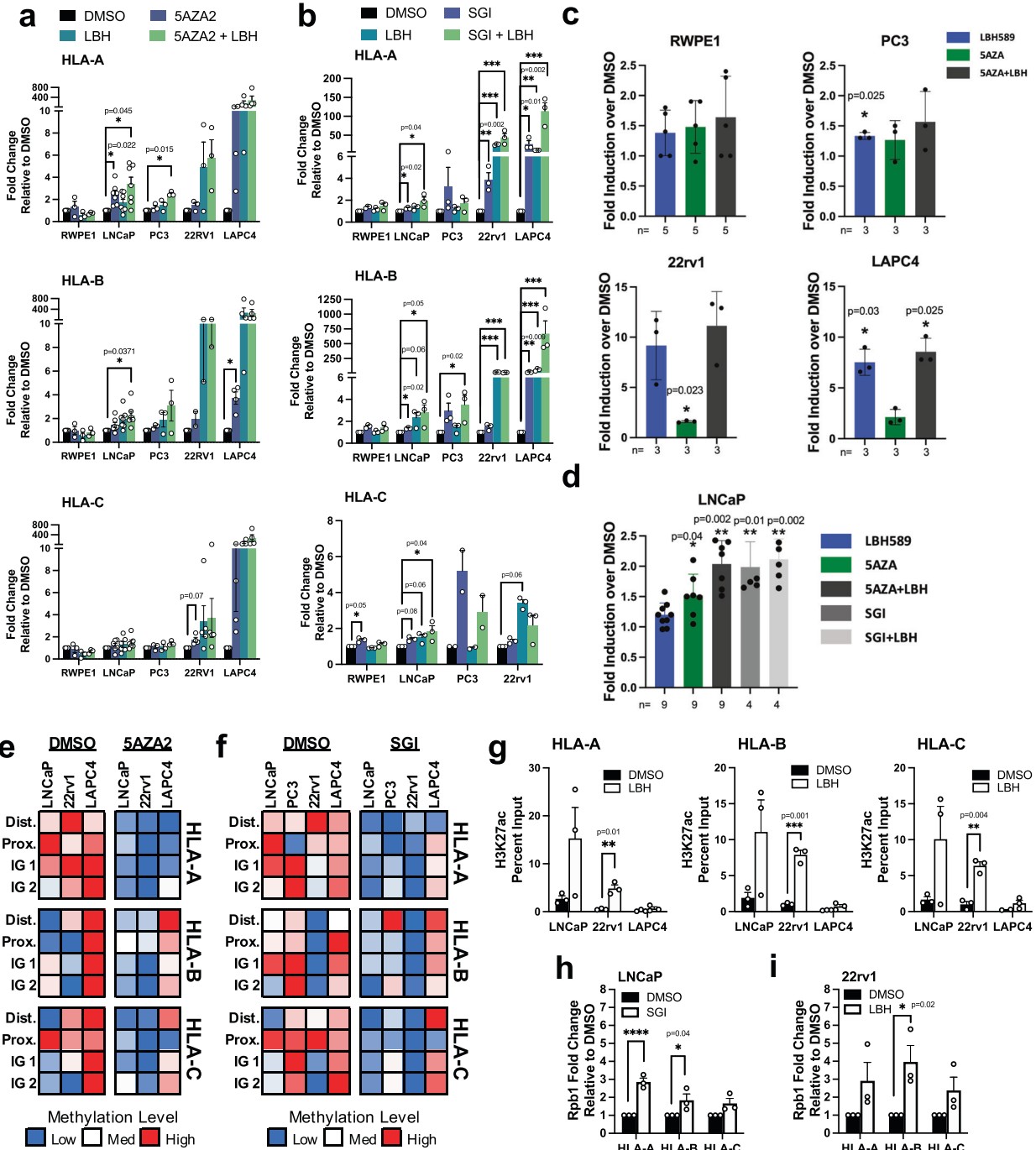

**Fig. 4 Pharmacological inhibition of DNMTs and HDACs induces HLA-I gene and protein expression.** Gene expression changes in HLA-A, HLA-B, and HLA-C in response to **a** 5AZA2 and LBH, with error bars representing SEM; n = 3 RWPE1, PC3, 22rv1 HLA-A; n = 4 LAPC4; LNCaP n = 5 HLA-A; n = 7 HLA-B, n = 6 HLA-C; n = 2 22rv1 HLA-B; n = 5 22rv1 HLA-C, except 5AZA2 n = 4 or **b** SGI and LBH alone and in combination. Data is from 3 independent experiments and shown as fold change relative to DMSO-treated cells. LAPC4 was excluded for *HLA-C* fold change analysis because *HLA-C* was undetectable in DMSO-treated LAPC4 cells. Error bars represent SEM. **c** Induction of HLA-I protein expression on the cell surface in indicated cell lines in response to 5AZA2 and LBH treatment alone and in combination. **d** Induction of HLA-I protein expression on the cell surface of LNCaP cells in response to 5AZA2, SGI, or LBH alone and in combination. Flow cytometry data in (**c**) and (**d**) is expressed as fold-over induction of Median Fluorescent Intensity normalized to DMSO-treated cells from at least three independent experiments. Mean +/− SD. **e** Heat map showing methylation levels in DMSO or 5AZA2 treated cells in four regions of the HLA-I genes from three independent experiments. **f** Heat map showing methylation levels in DMSO or SGI treated cells in four regions of the HLA-I genes from two independent experiments. **g** ChIP using antibodies to H3K27ac and H3K27me3 in LNCaP, 22rv1, and LACP4 cells treated with DMSO or LBH. **h** ChIP using an antibody to RPB1 in LNCaP cells treated with DMSO or SGI. **i** ChIP using an antibody to RPB1 in 22rv1 cells treated with DMSO or LBH. All ChIP data is representative of three independent experiments. For heat maps: Each heat map row was generated independent of other rows. Comparisons can be made between cell lines and treatment groups for each gene region individually, but not between different gene regions.

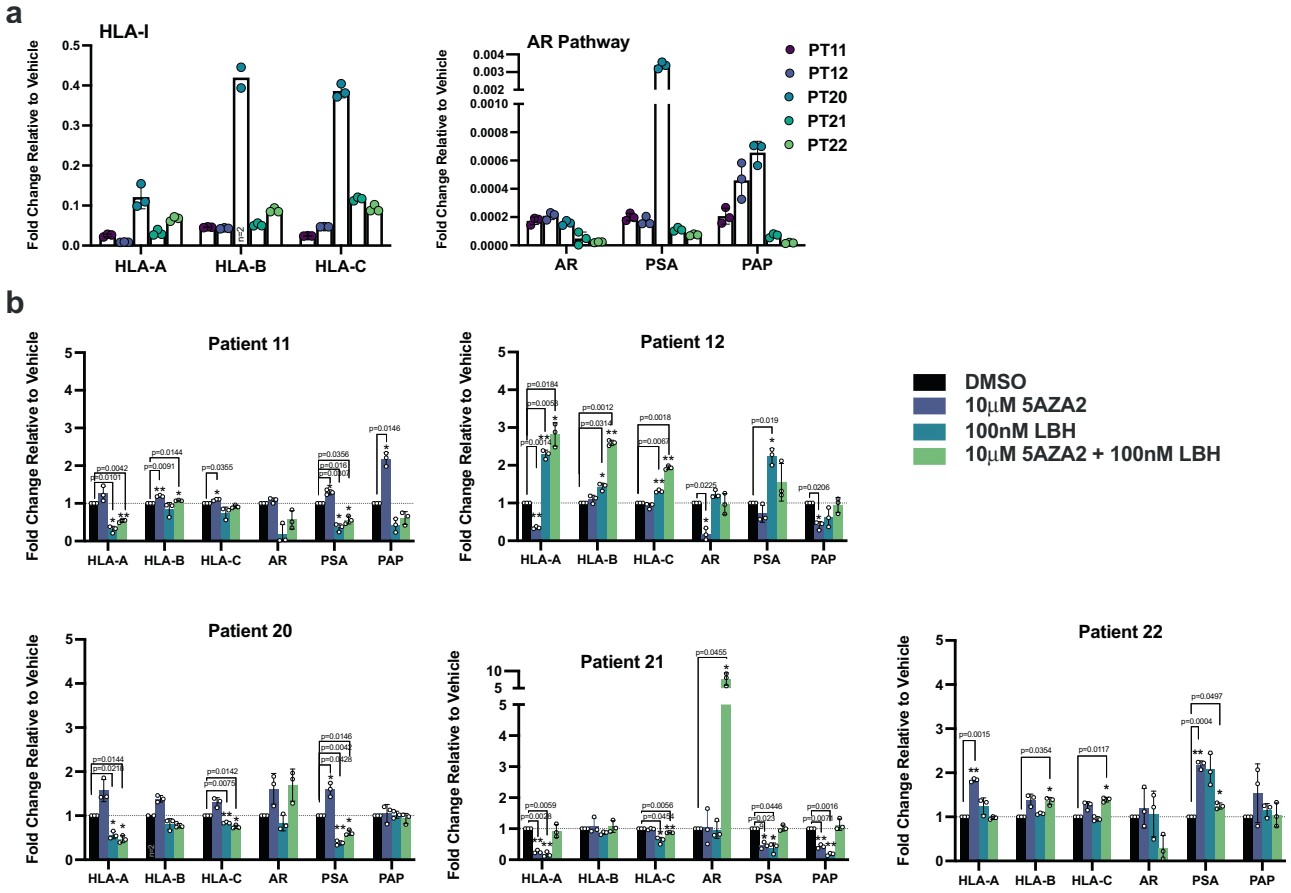

**Fig. 5 HLA-I expression and response to DNMT and HDAC inhibition ex vivo. a** Gene expression analysis of the HLA-I genes and prostate-specific elements of the AR pathway in prostate tumor tissue cultured ex vivo. **b** Gene expression analysis of induction of HLA-I and AR pathway elements in ex vivo tissue treated with DMSO, 5AZA2, LBH, or 5AZA2 + LBH; $n = 3$. Error bars represent SD.

of SGI or 5AZA2 and LBH and then co-cultured with splenocytes from vaccinated or unvaccinated mice (Fig. 6a). As an additional control, an anti-HLA-I blocking antibody was used to block HLA-I at the LNCaP cell surface. After co-culture, T-cell activation markers were measured by flow cytometry in T-cells from each co-culture treatment condition (Fig. 6b). We found that PSMA$_{27-38}$ tetramer-positive (PSMA$^+$) CD8$^+$ T-cells that were co-cultured with LNCaP cells treated with any DNMT or HDAC inhibitor increased in frequency and expressed increased levels of activation markers CD69 and LFA-1 compared to those co-cultured with DMSO-treated LNCaP cells. These T-cells also expressed increased levels of granzyme B and interferon-γ (IFN-γ), markers indicative of T-cell stimulation and differentiation into cytotoxic T-cells. In addition, the population of T-cells expressing CD107 (LAMP1), a marker of T-cell degranulation, was increased in the T-cells co-cultured with LNCaP cells treated with 5AZA2, SGI, and/or LBH.

The percent of PSMA$^+$ CD8$^+$ T-cells present after co-culture with LNCaP cells in each treatment condition is shown in Fig. 6c. Treatment of LNCaP cells with SGI and LBH in combination significantly increased the percent of PSMA$^+$CD8$^+$ T-cells after co-culture with LNCaP cells at a 2:1 or 1:1 T-cell effector to tumor target (E:T) ratio SGI alone was able to significantly increase the percent of PSMA$^+$CD8$^+$ T-cells after co-culture with LNCaP cells at a 2:1 or 1:1 T-cell effector to tumor target ratio. The percent of PSMA$^+$ CD8$^+$ T-cells present increased after co-culture with LNCaP cells treated with 5AZA2 and LBH in combination, though this did not reach statistical significance. Treatment of LNCaP cells with HLA-I blocking antibody was able to ablate these effects. No significant change in the percent of

PSMA$^+$ T-cells was seen when LNCaP cells were co-cultured with T-cells from unvaccinated mice. This study confirms a clear functional role for HLA-I induction by DNMT and HDAC inhibition and suggests the utility of HLA-I re-expression for vaccine-based immunotherapies that rely on functional MHC-I expression on tumor cells.

**HLA-I is methylated in prostate cancer circulating tumor cells with low HLA-I protein expression.** Circulating tumor cells (CTCs), which are thought to have metastatic potential, offer a window into the biology of solid tumors and have been validated as prognostic biomarkers in prostate cancer[38–40]. Since HLA-I gene expression was significantly reduced in metastatic lesions compared to primary tumors, we wanted to determine whether HLA-I downregulation is conserved in CTCs as part of the metastatic process. In addition, our in vitro and ex vivo data suggest patients have highly diverse responses to epigenetic therapy in the context of HLA-I re-expression and that only certain subsets of patients may benefit from specific therapies. Therefore, developing an epigenetic-based biomarker would be useful to determine which patients harbor epigenetic modifications in HLA-I and may benefit from epigenetic therapy.

We first characterized HLA-I protein expression in CTCs, as this has not yet been reported in the literature. We isolated prostate cancer CTCs from whole blood and stained with Hoechst and antibodies targeting cytokeratin to identify cells of epithelial origin, a panel of white blood cell markers to identify white blood cells, and HLA-I. We gathered separate cohorts to stain CTCs for both extracellular HLA-I and intracellular HLA-I and compared

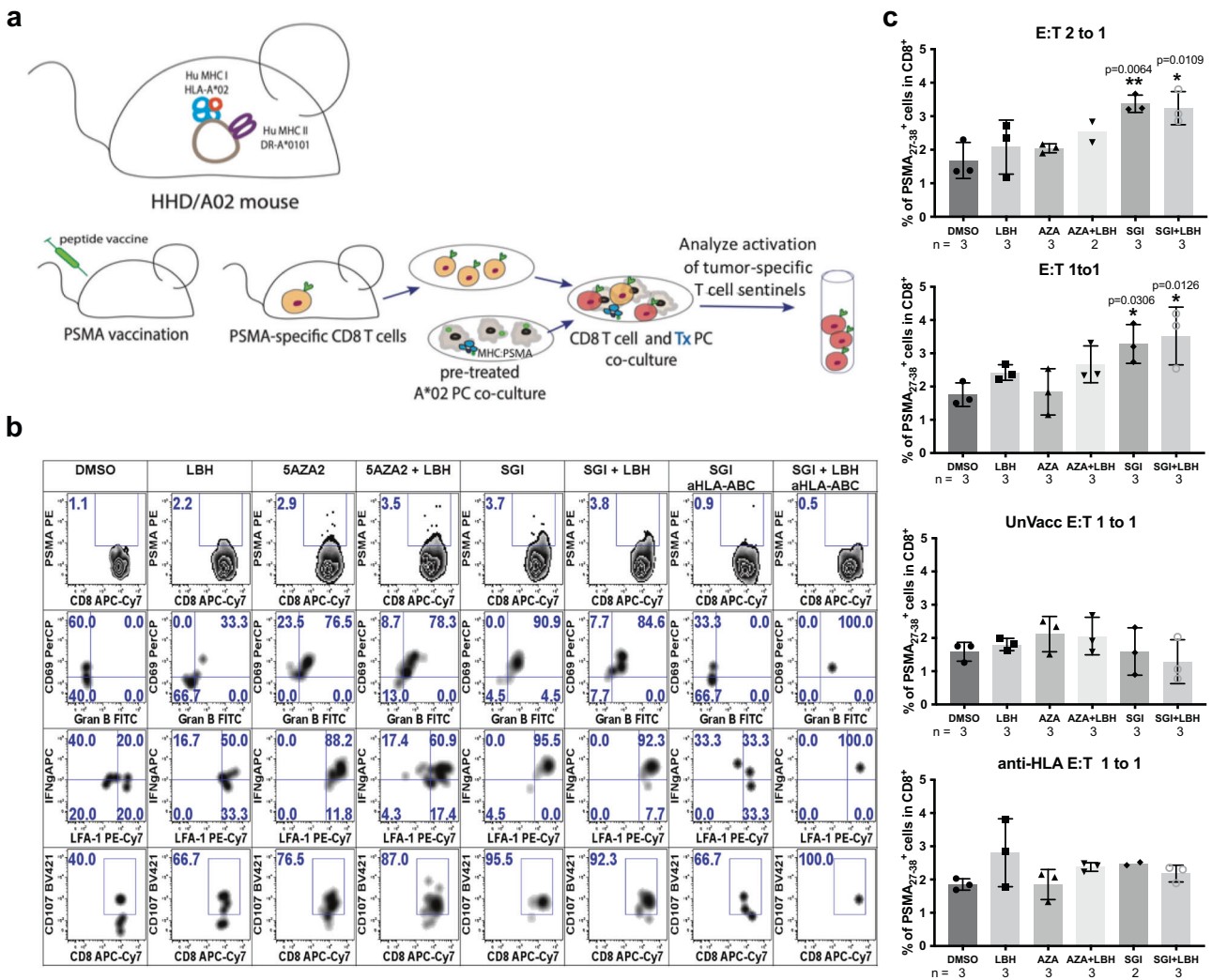

**Fig. 6 DNMT and HDAC inhibition in tumor cells increase co-cultured T-cell activation. a** Vaccination and co-culture scheme to analyze PSMA$_{27-38}$-specific T-cell response to LNCaP cells treated with epigenetic modifying agents. **b** Plots represent the rare-event cell analysis of PSMA$_{27-38}$-specific T-cells to assess the cytolytic features. Top row shows PSMA$_{27-38}$ tetramer binding on the total CD8$^+$ splenocyte subset. PSMA$_{27-38}$ tetramer$^+$ cells are projected in rows 2–4 and gate frequencies are expressed as a percentage of total PSMA$_{27-38}$ CD8$^+$ T-cells. Plots represent expression of activation markers CD69, LFA-1, IFNγ, granzyme B, and CD107 (LAMP1). **c** Graphs represent the frequency of PSMA$_{27-38}$ pentamer positive cells within the total CD8$^+$ T-cell population after co-culture with LNCaP cells treated with epigenetic modifying agents. Splenocytes were co-cultured with LNCaPs in ratios of 2:1 and 1:1 PSMA$_{27-38}$ tetramer$^+$ CD8$^+$ T-cell Effector to tumor Target (E:T). In control conditions, splenocytes from unvaccinated mice were co-cultured with LNCaPs treated with epigenetic modifying agents or HLA-I as blocked by an anti-HLA-A, B, C blocking antibody. Error bars represent SD.

the expression of HLA-I in individual CTCs to the median expression of HLA-I in the matched WBC populations (Fig. 7a). We found that intracellular HLA-I expression in CTCs is heterogenous, but 7 out of 8 patients had at least one CTC with expression below the median WBC level of expression. Half of the patients had intracellular HLA-I expression below the median WBC level in all CTCs detected. For the extracellular expression cohort, all 8 samples had CTCs with extracellular expression of HLA-I below the level of median WBC expression and 7/8 patients had the majority of CTCs fall below the WBC median expression level. A representative image of a CTC and WBC is shown in Fig. 7b.

We then evaluated whether CTCs with low HLA-I protein expression also harbored increased levels of HLA-I DNA methylation. To do this, we utilized a semi-automated single-cell aspiration system to individually aspirate CTCs with high or low HLA-I protein expression[41]. EpCAM-positive CTCs were isolated and stained with Hoechst and antibodies to HLA-ABC,

EpCAM, and markers of WBCs[42,43]. Stained cells were seeded onto a microwell array for aspiration (Fig. S8a). CTCs were then aspirated based on relative levels of HLA-ABC binding determined by fluorescence microscopy for each patient sample. A representative example image of a cell in each of these populations is presented in Fig. 7c. Groups of ~10–15 HLA-I positive or HLA-I negative CTCs were collected. We confirmed HLA-I expression differences in the identified populations by analyzing the mean fluorescent intensity (MFI) of HLA-ABC binding in each group (Fig. 7d). HLA-I expression in the HLA-I negative CTC populations was significantly lower than the HLA-I positive CTC populations in all patients and significantly lower than matched WBC populations where available (3 out of 4 patients). Of note, HLA-I expression in the HLA-I positive CTC groups was lower than matched WBCs. Though the differences between the HLA-I positive CTC groups and WBCs were not statistically significant, this lower level of HLA-I expression may be functionally relevant and epigenetically regulated. A negative

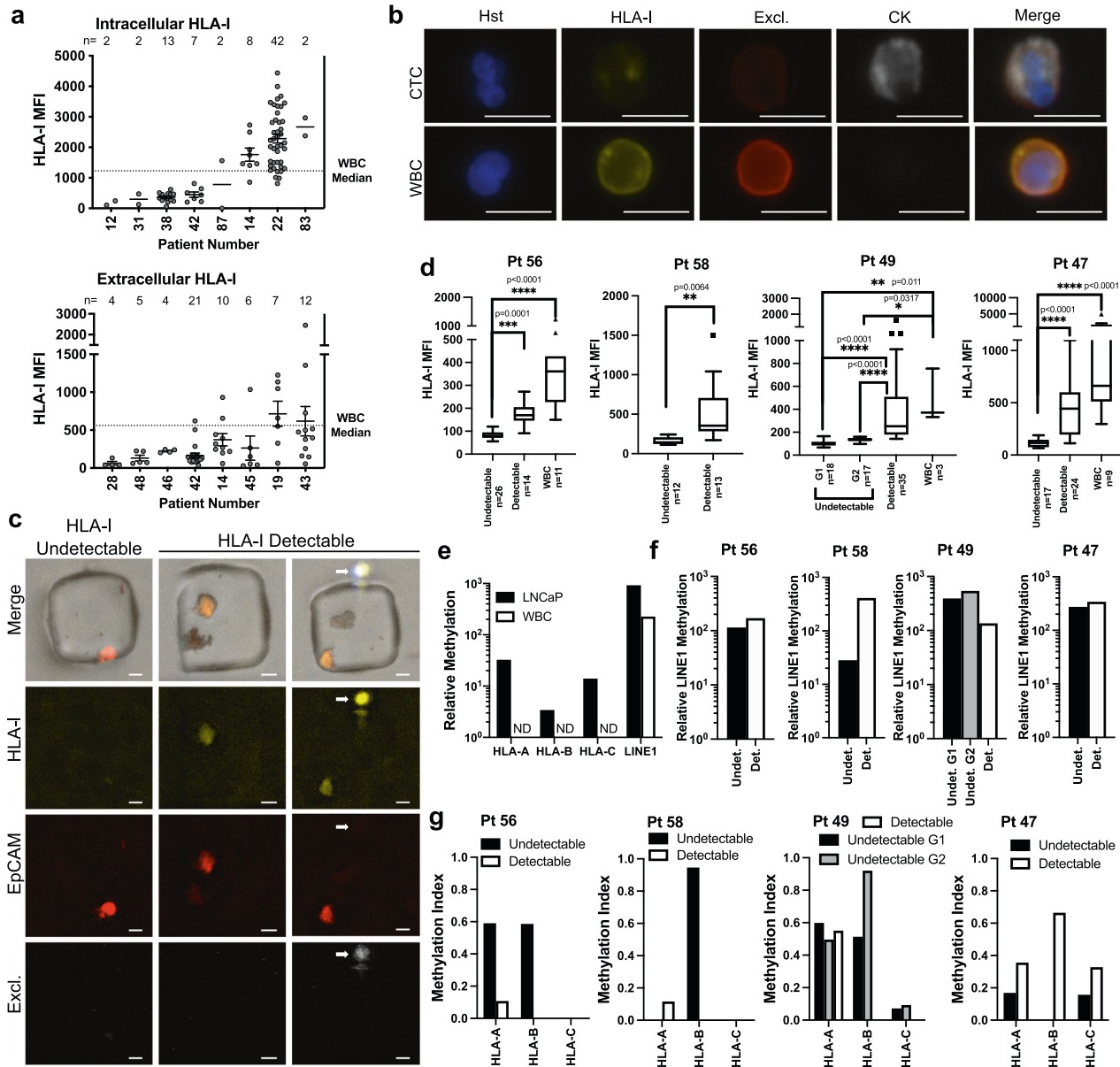

**Fig. 7 HLA-I expression and methylation in circulating tumor cells. a** Intra- and extracellular HLA-I protein expression in circulating tumor cells from two cohorts of 8 patients with prostate cancer. Dotted line represents median HLA-I expression in all patient matched white blood cells (WBCs). **b** Representative ×40 images of a CTC with low HLA-I expression and a WBC near the median expression from the total HLA-I group. Hoechst (Hst; blue; CD45/CD34/CD11b/CD27 (Excl; red); pan-cytokeratin (CK; white). Scale bars represent 10 µM. **c** Representative 10x images of HLA-I (yellow) undetectable and HLA-I detectable CTCs and a WBC (white arrow) in the single-cell aspirator microwells (Pt. 56) HLA-I (yellow), EpCAM (red), Excl. (white). Scale bars represent 10 µm. **d** Mean fluorescent intensity (MFI) of HLA-I in each CTC population and matched WBCs (where available). Statistical analysis was done by one-way ANOVA with the Kruskal–Wallis test, and Welch's t-test for Pt 58. Approximately 10–15 CTCs were chosen to be aspirated from each group. **e** HLA-I methylation in 10 WBCs and 10 LNCaP cells purified by single-cell aspiration. **f** Relative LINE1 methylation in groups of ~10–15 CTCs purified by single-cell aspiration. **g** HLA-I methylation in HLA-I detectable and undetectable groups of ~10–15 CTCs purified by single-cell aspiration represented with a Methylation Index relative to methylation in LNCaP cells.

control group of 10 WBCs as well as a positive control group of 10 LNCaP cells were collected by single-cell aspiration for comparison.

Methylated DNA was enriched using the SEEMLIS (Semi-Automated ESP(Exclusion-Based-Sample-Preparation) Enrichment of Methylated-DNA from Low Input Samples) method of MBD2 protein-based enrichment of enzymatically digested DNA from the collected groups of CTCs and controls[44]. Enriched methylated DNA was subjected to qPCR with primers targeting HLA-I genes and *LINE1*, a control for successful enrichment of

methylated DNA (Fig. S8b). We were able to successfully detect HLA-I methylation from 10 aspirated LNCaP cells, while no methylation was detected in the WBC group (Fig. 7e). *LINE1* methylation was detected in all patient samples and controls (Fig. 7e, f). We were able to detect HLA-I methylation in all four patient samples, though the pattern of detection and association with protein expression varied (Figs. 7g and S8b).

Patients 568, 588, and 490 had higher methylation in the HLA-I genes in the HLA-I protein negative CTC populations than the HLA-I protein positive CTC populations. Levels of methylation

were between 50% and 100% of the level of methylation in the LNCaP control sample for *HLA-A* and *HLA-B*, when detected, in these patients. *HLA-C* methylation levels were lower, but present in the HLA-I protein negative populations in patient 490. In contrast, patient 487 had detectable HLA-I methylation in both groups of CTCs, with higher levels detected in the HLA-I positive population. *HLA-A* was also detected in the HLA-I protein-positive populations at low levels in patient 568 and 588 and at higher levels in patient 490. As mentioned above and shown in Fig. 7d, the HLA-I expression level in the HLA-I positive CTC groups tended to be lower than the matched WBC expression, suggesting that low, non-negative HLA-I expression may still be epigenetically regulated in some patients. These patients may still benefit from epigenetic therapy to express HLA-I to levels closer to that of WBCs. Overall, this experiment represents the first analysis of HLA-I methylation in prostate cancer CTCs and is a starting point for future studies on methylated DNA biomarkers for prostate cancer therapies.

## Discussion

The evidence presented here demonstrates that epigenetic mechanisms regulate expression of HLA-I genes in human prostate cancer. While it has been known for decades that HLA-I is downregulated at the protein level in prostate cancer, there have been limited investigations into the molecular underpinnings of this phenomenon, especially as it relates to epigenetic regulation of these genes. To address this lack of understanding, we have described the HLA-I methylome in prostate cancer patient samples and cell lines and confirmed the presence of repressive histone modifications in cell line models. To the best of our knowledge, we have also for the first time measured HLA-I protein expression in circulating tumor cells (CTCs) from patients with prostate cancer and found that low HLA-I expression in CTCs was associated with increased levels of HLA-I DNA methylation.

Our investigations into the regulation of class I HLA genes in prostate cancer revealed frequent downregulation of HLA-I gene expression in metastatic tumors. The striking decrease in HLA-I expression and the presence of *HLA-A* DNA methylation signatures in metastatic lesions identified in this study implies a possible role for epigenetic loss of HLA-I expression in progression to metastasis. This idea is supported by previous findings showing that promoter DNA methylation increases during progression and that epigenetic mechanisms are important drivers of prostate cancer progression[17,45,46]. Whether the epigenetic alterations in HLA-I genes are a driver or passenger in the metastatic cascade will need to be further studied, but our data suggest that immune evasion through epigenetic downregulation of HLA-I is a frequent event in metastatic prostate cancer.

This study is in accordance with previous work showing HDAC inhibition upregulates HLA-I gene and protein levels in LNCaP cells[14], and gene expression levels in PC3 and Du145 cells[14,47]. Our work expands on these studies by examining the basal epigenetic signatures as well as changes in those signatures in response to HDAC and DNMT inhibition in multiple cell lines. We strengthen this idea further by showing biological relevance for HLA-I induction by DNMT and HDAC inhibition in ex vivo tissue samples. Importantly, we also showed that HLA-I gene and protein expression can be functionally induced on tumor cells by DNMT and HDAC inhibition, leading to increased activation of co-cultured T-cells from mice vaccinated with PSMA-peptide. This study demonstrated the potential power of combining therapeutic re-expression of epigenetically silenced HLA-I with an MHC-I-dependent immunotherapy.

A previous study from our lab found DNMT and/or HDAC inhibition induces expression of CTAs and another study found

APM molecules were upregulated with HDAC inhibition[42,47]. Our current study supports a wider role for epigenetics in regulating antigen presentation by also downregulating the HLA-I genes themselves. Inhibition of DNMT and HDACs likely affects many cellular pathways in addition to HLA-I genes, CTAs, and APM, leading to the phenotypes we observed in this study. In line with this, our ex vivo gene expression analysis suggests a role for APM and B2M epigenetic regulation in prostate cancer patients, which merits further study. A recent study found inhibition of BET bromodomain-containing proteins, which are readers of histone acetylation, led to increased HLA-I protein expression and immunogenicity in vivo, supporting the important role we have found for histone modifications in HLA-I regulation[48]. Further investigation into the contributions of other affected cellular pathways is needed to fully understand this phenomenon.

DNMT and HDAC inhibitors have been explored for their possible therapeutic efficacy and numerous clinical trials are ongoing for single or combination uses, including trials with 5AZA2, SGI, and LBH[49]. Recently, the first epigenetic therapy for solid tumors, a small molecule drug targeting EZH2, was approved by the FDA for use in epithelioid carcinoma[50,51]. Previously, the only FDA approved uses for drugs targeting epigenetic modifying proteins were for hematologic malignancies. The limited success of epigenetic therapies in solid cancers may be due to the heterogeneity in the epigenetic signatures and responses to therapies as evidenced in this study[23]. Further preclinical and clinical-translational studies are needed to investigate the effect of epigenetic therapies on various tumor microenvironment elements and also beyond the tumor microenvironment, including effects on local and systemic immune function to fully assess translational utility of epigenetic therapy in solid tumors. In addition, failure of immunotherapies has been attributed to lack of immunogenicity of tumor cells as well as the inability to monitor treatment response in solid tumors[24]. We anticipate that the ability to identify patients likely to respond to therapeutic re-expression of HLA-I and combination immunotherapy by monitoring epigenetic signatures in circulation may alleviate some of these challenges. This study is the first to measure HLA-I expression and matched methylation in CTCs. Further investigation into how HLA-I methylation in circulation can inform on treatment options or outcomes is certainly warranted and is a goal of our future studies. Overall, this study has implicated epigenetic mechanisms in the regulation of HLA-I in prostate cancer and points to epigenetic therapy as a promising option for enhancing the immune response in prostate tumors.

## Methods

**Analysis of public data sets**. Genomic alteration data was accessed and analyzed in cBioPortal (RRID: SCR_014555)[52,53]. TCGA-PRAD data was accessed and downloaded through UCSC Xena[54]. Data from Taylor, et al. was accessed, analyzed, and downloaded through cBioPortal[27]. ATAC-seq data from Corces, et al. was accessed, analyzed, and downloaded through UCSC Xena[54,55]. Methylation beta values and matched gene expression values were accessed through Wanderer[56]. Probe information is listed in Supplementary Data S4. Prism 8 (GraphPad Prism, RRID: SCR_002798) was used for correlation analyses. Z-scores for gene expression and ATAC-seq were calculated with the following formula:

$$Z = \frac{\chi - \mu}{\sigma} \qquad (1)$$

where $\chi$ is the tumor or metastasis gene expression value, $\mu$ is the normal sample population mean, and $\sigma$ is the normal sample population standard deviation. In the ATAC-seq data set, z-scores were calculated as compared to the tumor population, where $\chi$ is the tumor gene or ATAC-seq expression value, $\mu$ is the tumor sample population mean, and $\sigma$ is the tumor sample population standard deviation.

**COMPARE-MS**. Patient samples used in this analysis were previously described[30,33]. COMPARE-MS was performed as previously described[30,31,33]. Briefly, DNA was digested with AluI and HhaI restriction enzymes (New England Biolabs, Cat# R0137L and Cat# R0139L) and enriched with MBD2-MBD protein (Takara Bio, Cat# 631962). Enriched DNA was subjected to qPCR with primers

targeting *HLA-A* (Fwd: TCTGCGGGGAGAAGCAAG; Rev: GGGGACACGG ATGTGAAGAAA). Methylation index was calculated by normalizing Ct values from samples to controls (enzymatically methylated white blood cell DNA) to generate a Methylation Index from a range of 0.0–1.0.

**Cell lines and cell culture.** LAPC4 (ATCC, Cat# CRL-13009, RRID: CVCL_4744) were maintained in DMEM Medium (Corning) supplemented with 20% fetal bovine serum (FBS) (Gibco, Cat# 10437028), 1% sodium pyruvate (Corning, Cat# MT25000CI), 0.5% beta-mercaptoethanol, and 1% penicillin-streptomycin (HyClone, Cat# SV30010). LAPC4 cells were cultured in poly-d-lysine coated flasks and/or plates (BioCoat flasks: Corning, Cat# 0877260; 6-well plates: Sigma-Aldrich, Cat# Z720798-20EA). RWPE1 (ATCC Cat# CRL-11609, RRID: CVCL_3791, LNCaP (ATCC, Cat# CRL-1740, RRID: CVCL_1379), 22rv1 (ATCC, Cat# CRL-2505, RRID: CVCL_1045), and PC3 (ATCC, Cat# CRL-1435, RRID: CVCL_0035) cells were maintained in RPMI 1640 Medium (Corning, Cat# MT10040CV) supplemented with 10% FBS, 1% sodium pyruvate, 1% penicillin-streptomycin, 1% non-essential amino acids (HyClone, Cat# SH30238.01), and 0.1% beta-mercaptoethanol. LCL (HCC2218-BL, ATCC, Cat# CRL-2363, RRID: CVCL_1264) cells were grown in suspension in RPMI 1640 Medium supplemented with 10% FBS and 1% penicillin-streptomycin. LCL, RWPE1, LNCaP, 22rv1, and PC3 were cultured in tissue culture treated flasks and/or plates (Flasks: Corning, Cat# 07202000; Plates: Thermo Fisher Scientific, Cat# 087721G). All cell lines were used in experiments within 30 passages from thaw. RWPE1, LNCaP, PC3, and LAPC4 were authenticated by short tandem repeat and tested for mycoplasma by PCR in 2017 at the TRIP Laboratory at the Department of Pathology, University of Wisconsin.

**Ex vivo culture of prostate tissue.** Human prostate tissues were obtained from patients undergoing radical prostatectomy at the University of Wisconsin-Madison. All patients were consented in writing under an Institutional Review Board (IRB) protocol #20130653. Research has been performed in accordance with the Declaration of Helsinki. All the laboratory investigators were blinded to clinical information. Absorbable gelatin sponges (Ethicon, Cat# 1973) were cut into pieces to fit in a 24-well tissue culture plate. Sponges were soaked in Ham's F-12 media (Fisher Scientific, Cat# SH3002601) supplemented with 0.25 units/ml regular insulin (Sigma-Aldrich, Cat# I9278-5ML), 1 µg/mL hydrocortisone (Sigma-Aldrich, Cat# H0888-1g), 5 µg/mL human transferrin (Sigma-Aldrich, Cat# T8158-100mg), 2.7 mg/ml dextrose, 0.1 nM non-essential amino acids (HyClone, Cat# SH30238.01), 100 units/ml and 100 µg/mL Penicillin/Streptomycin, respectively (HyClone, Cat# SV30010), 2 mM L-glutamine (Corning, Cat# 25-005-CI), 25 µg/mL bovine pituitary extract (Life Technologies, Cat# 13028014), and 1% fetal bovine serum (FBS) (Gibco, Cat# 10437028) until fully saturated. Each core was cut into ~1 mm² by 1 mm² cubes. Tissue was placed on the sponges and cultured for up to 4 days at 37 °C at 5% $CO_2$ and 500 µL media was replaced daily.

**Immunoblotting.** Whole-cell lysates were collected from adherent cells by scraping into RIPA buffer after washing with cold PBS. Whole-cell lysates were separated by SDS-PAGE and transferred onto nitrocellulose membrane. Membranes were blocked with SuperBlock blocking buffer (Thermo Scientific, Cat# 37515). Membranes were probed with primary antibodies diluted in 3% BSA in TBS plus 0.1% Tween-20 at 4 °C overnight followed by incubation with HRP-linked secondary antibody (BioLegend, Cat# 405306, RRID:AB_315009, 1:5000) at RT for 1 h and visualization by chemiluminescence. Primary antibodies: HLA-I clone W6/32 (BioLegend, Cat# 311412, RRID:AB_493132, 1:1000), α-tubulin (BioLegend, Cat# 627901, RRID:AB_439760, 1:1000).

**Flow cytometry analysis of cell lines.** Cells were stained with Ghost Dye Violet 510 (Tonbo Biosciences, Cat# 13-0870) to identify viable cells and PE-Cy7 conjugated anti-HLA-ABC antibody (BioLegend, Cat# 311430, RRID:AB_2561617). Intracellular staining for PSMB8 (LMP7)-PE (Abcam, Cat#EPR14482), Calreticulin-AlexaFluor647 (MBL International, Cat#K0136-4; RRID:AB_592808), and TAP-1-FITC (Abcam, Cat#EPR3924; RRID:AB_2819061) protein expression was performed following manufacturer's protocol with BD Cytofix/Cytoperm kit (BD Biosciences, Cat#554714; RRID:AB_2869008). Samples were acquired on an LSR II instrument and data analyzed by the FlowJo software v9.9.6 (FlowJo, RRID: SCR_008520). Median Fluorescent Intensity was analyzed on gated live, single cells.

**MBD2-MBD enrichment of methylated DNA from cell lines.** Genomic DNA was isolated from cells using the AllPrep RNA/DNA Mini Kit (Qiagen, Cat# 80204) according to manufacturer's instructions. DNA was quantified by a NanoDrop 1000 spectrophotometer and 1 µg DNA was sheared by sonication to an average size of around 200 bp. Methylated DNA was enriched from sheared genomic DNA using the EpiXplore Methylated DNA Enrichment Kit (Takara Bio, Cat# 631962) according to manufacturer's instructions. Enrichment was measured by qRT-PCR using primers designed to various regions of *HLA-A*, *HLA-B*, and *HLA-C* (Integrated DNA Technologies). Primers are listed in Supplementary Data S5.

**Chromatin immunoprecipitation.** Chromatin immunoprecipitation (ChIP) was performed according to manufacturer's instructions using the SimpleChIP Enzymatic Chromatin IP Kit with Magnetic Beads (Cell Signaling Technology, Cat# 9003S). Immunoprecipitation was performed using the following antibodies from Cell Signaling Technology: Histone H3 (Clone D2B12, Cat# 4620S, RRID: AB_1904005), H3K27ac (Clone D5E4, Cat# 8173S, RRID: AB_10949503), H3K27me3 (Clone C36B11, Cat# 9733S, RRID: AB_2616029), and IgG (Cat# 2729S, RRID: AB_1031062). DNA was then analyzed by qPCR using primers designed to target *HLA-A*, *HLA-B*, and *HLA-C* (Integrated DNA Technologies). Primers are listed in Supplementary Data S5. The H3K27ac signature in GM12878 determined by ChIP-seq from the ENCODE consortium[35] was accessed in the UCSC Genome Browser (RRID: SCR_005780) to aid in primer design.

**Epigenetic drug treatments of cell lines and ex vivo tissue.** 5-Aza-2′-deoxycytidine (5AZA2) (Sigma-Aldrich, Cat# A3656-5MG), Panobinostat (LBH, LBH589) (Selleckchem, Cat# S1030), and SGI-110 (SGI) (Astex Pharmaceuticals) were dissolved in DMSO and stored at −80 °C in aliquots. Cells were treated with 10 µM 5AZA2, 1 µM SGI, or DMSO for 72 h. 10 nM or 100 nM LBH was added for the last 24 h after 48 h of 5AZA2 or SGI treatment for combination treatments.

**Gene expression analysis in cell lines.** Total RNA was isolated from cells using the AllPrep RNA/DNA Mini Kit (Qiagen, Cat# 80204) according to manufacturer's instructions. RNA was quantified by a NanoDrop 1000 spectrophotometer and 1 µg total RNA was reverse transcribed using the High Capacity RNA-to-cDNA kit (Thermo Fisher Scientific, Cat# 4388950). cDNA was diluted 10x and 5 µL was used per reaction for qPCR. Pre-designed TaqMan probes (Thermo Fisher) for *HLA-A* (Hs01058806_g1), *HLA-B* (Hs00818803_g1), and *HLA-C* (Hs00740298), and *RPLP0* (4333761F) were used with iTaq Universal Probes Supermix (BioRad, Cat# 1725135). Gene expression was determined using the delta-delta-Ct method after normalization of each gene to housekeeping gene, *RPLP0 (P0)*.

**Gene expression analysis in epigenetic drug treated ex vivo tissue and 5AZA2/LBH-treated cell lines.** Total RNA was isolated from cells using Rneasy Mini Kit (Qiagen, Cat# 74106) according to manufacturer's protocol. Total RNA was isolated from ex vivo tissue using the Aurum Total RNA Fatty and Fibrous Tissue Kit (BioRad, Cat# 7326830) according to the manufacturer's protocol. RNA was quantified by a NanoDrop 1000 spectrophotometer and 1 µg total RNA was reverse transcribed using iScript Reverse Transcription Supermix (Bio-Rad, Cat# 1708841). 1 µL of the cDNA synthesis reaction was used to perform qPCR using SsoAdvanced Universal SYBR Green Supermix (BioRad, Cat# 1725274) according to the manufacturer's protocol. *HLA-A*, *HLA-B*, *HLA-C*, and *RPLP0* primers are listed in Supplementary Data S5. Gene expression was determined using the delta-delta-Ct method after normalization of each gene to housekeeping gene, *RPLP0 (P0)*.

**Peptide vaccinations and T-cell co-culture.** PSMA-specific CD8+ T-cells were generated by PSMA$_{27-38}$ peptide vaccination of HHD transgenic humanized mice expressing human HLA-I A*02. The HHD mice were a generous gift from Professor Francois Lemonnier at the Pasteur Institute, Paris[57]. Research animals were maintained and experiments were performed in accordance with institutional guidelines overseen by the Institutional Animal Care and Use Committee of the University of Wisconsin. Mice were given once weekly subcutaneous injections of 100 µg synthetic PSMA peptide (VLAGGFFLL) (ProImmune, Oxford, UK) in 100 µL CFA (Thermo Fisher Scientific, Cat# NC0916022) for the first injection or IFA vehicle (Sigma-Aldrich, Cat# AR002) for subsequent injections. Splenocytes were harvested 1 week after last immunization and the number of live PSMA$_{27-38}$-specific CD8+ splenocytes was determined by flow cytometry analysis following staining with GhostDye Violet 510 (Tonbo Biosciences, Cat# 13-0870), anti-mouse CD8 antibody (Tonbo Biosciences, Cat# 25-0081, RRID:AB_2621623) and Pro5 → PSMA$_{27-38}$ A*02:01 MHC I pentamer (ProImmune, Oxford, UK). PSMA vaccinated splenocytes were then co-cultured with LNCaP cells that were pre-treated with DMSO vehicle or 10 µM 5AZA2 or 1 µM SGI for 72 h and/or 10 nM LBH for the last 24 h in RPMI media supplemented with 10% FBS. In control co-culture wells, LNCaP cells were treated with 5 µg of purified anti-HLA-A,B,C blocking antibody (clone W6/32) (BioLegend, Cat# 311412, RRID:AB_493132) prior to adding splenocytes. Cells were co-cultured for 72 h and Golgi-stop (BD Biosciences, Cat# 554724) was added for the last 4 h of culture, following the manufacturer's protocol. Cells were then harvested and subjected to labeling with Ghost Dye Violet 510 and surface markers: CD8 (Tonbo Biosciences, Cat# 25-0081, RRID:AB_2621623), Pro5 → PSMA$_{27-38}$ A*02:01 MHC I pentamer (ProImmune), CD69 (BD Pharmigen, Cat# 551113, RRID:AB_394051), LFA-I (BD Biosciences, Cat# 558191, RRID:AB_397055), CD107 (BD Biosciences, Cat# 564347, RRID:AB_2738760), and CD16/CD32 Fc Block 2.4G2 (BD Biosciences Cat#553142) followed by fixation and permeabilization with BD Cytofix/Cytoperm (Thermo Fisher Scientific, Cat# 554714) according to the manufacturer's protocol and intracellular staining with antibodies against murine IFNγ (Tonbo Biosciences, Cat# 20-7311, RRID:AB_2621616) and Granzyme B (BD Biosciences, Cat# 560211, RRID:AB_1645488). Cells were then washed and acquired on an LSR II Fortessa or an Attune NxT instrument followed by data analysis by the FlowJo software v9.9.6

(FlowJo, RRID: SCR_008520). Gating controls included the fluorescent minus one (FMO) strategy.

**CTC capture, imaging, and analysis**. Blood samples were collected from prostate cancer patients after receiving written informed consent under a protocol approved by the Institutional Review Board at the University of Wisconsin-Madison (#2014-1214). Research has been performed in accordance with the Declaration of Helsinki. Patients were required to have histologically confirmed prostate adenocarcinoma, and documented metastases, as confirmed on computed tomography (CT) or bone scanning with technetium-99m-labed methylene diphosphonate. Peripheral blood samples, for analysis of CTCs, were obtained from eligible patients at the time of disease evaluation. All the laboratory investigators were blinded to clinical information when determining CTC results. CTCs were processed and stained as previously described in Sperger et al.[42]. Briefly, PBMCs were isolated from whole blood on Ficoll-Paque PLUS (GE Healthcare, Cat# 45-001-750) gradient and fixed with Cytofix Fixation Buffer (BD Biosciences, Cat# 554655). Fixed cells were incubated with paramagnetic particles (PMPs) (Dynabeads® FlowComp™ Flexi kit, Life Technologies, Cat# 11061D) coated with biotinylated anti-EpCAM antibody (R&D Systems, Cat# AF960, RRID: AB_355745). The Versatile Exclusion-based Rare Sample Analysis (VERSA) platform was used for enrichment and staining of CTCs[42,58,59]. PMP bound cells were isolated in the VERSA and stained with Hoechst 33342 (Thermo Fisher, Cat# PI62249) and antibodies to the proteins indicated in the corresponding figures, which are summarized in Supplementary Data S6. Pan-cytokeratin was conjugated to Alexa Fluor 790 using the Alexa Fluor 790 Antibody Labeling Kit (Life Technologies, Cat# A20189) according to manufacturer's instructions. All other antibodies were purchased pre-conjugated to the fluorophores listed in Supplementary Data S6. CD45, CD34, and CD11b were used on the same channel to serve as a white blood cell (WBC) "exclusion channel". CD14 and CD27 were included in addition to CD45, CD34, and CD11b in the WBC exclusion channel for the experiment measuring extracellular HLA-I in CTCs only. Cells were stained for extracellular markers at 4 °C for 30 min. Cells were permeabilized and stained for intracellular antibodies with BD Perm/Wash at 4 °C overnight (BD Biosciences, Cat# BDB554723). Cells were imaged in the VERSA at 10x magnification using NIS Elements AR Microscope Imaging Software (NIS-Elements, RRID:SCR_014329) and analyzed using NIS Elements analysis software. CTCs were defined as positive for Hoechst and pan-cytokeratin and negative for CD45/34/11b or CD45/34/11b/14/27. All other cells were considered part of the WBC population.

**Single-cell aspiration of CTCs for methylation analysis**. CTCs enriched using the VERSA platform as described above were stained in the VERSA with Hoechst 33342 (Thermo Fisher, Cat# PI62249) and antibodies to HLA-A,B,C, EpCAM, CD27, CD45, CD34, and CD11b. Fluorophores, catalog numbers, and other antibody information is summarized in Supplementary Data S6. Patient characteristics for single-cell aspiration CTC samples are summarized in Supplementary Data S7. Cells were then further enriched using a single-cell aspiration platform, SASCA, previously developed by Tokar et al.[39]. Briefly, cells were seeded into polydimethylsiloxane (PDMS) microwells mounted on a glass microscope slide. The microwell array was imaged on a Nikon Ti-E Eclipse inverted fluorescent microscope. CTCs were identified as EpCAM positive, exclusion (CD45/CD34/CD11b/CD27) negative cells, whereas WBCs were defined as EpCAM negative, exclusion positive cells. CTCs were further subdivided into groups based on HLA-I positivity compared to WBCs in the same sample. Target cells were aspirated from microwells and dispensed into a droplet of PBS in the EXTRA-CTMAN extraction plate (Gilson, Cat# 22100008) to proceed with DNA extraction. Microarray images were analyzed using NIS Elements AR Microscope Imaging Software (NIS-Elements, RRID:SCR_014329) to obtain HLA-I mean fluorescent intensity (MFI) values.

**DNA extraction from CTCs**. DNA extraction was performed as previously described[44]. DNA was extracted using a semi-automated Gilson PIPETMAX liquid handling robot enabled for exclusion-based sample preparation (ESP), termed EXTRACTMAX[60]. The robot added LiDS buffer (90 mM Tris-HCL, 500 mM lithium chloride, 1% Igepal CA-630, 10 mM EDTA, 1 mM dithiothreitol) and MagneSil Paramagnetic Particles (PMPs) (Promega, Cat# MD1441) resuspended in GTC buffer (10 mM Tris-HCl, 6 M guanidinium thiocyanate, 0.1% Igepal CA-630, pH 7.5) to the extraction microplate (Gilson, Cat# 22100008). The robot then adds cells in suspension to the well-containing LiDS, GTC, and MagneSil beads. Cells are mixed in the buffer by the robot. Cells were lysed and DNA was allowed to bind to MagneSil PMPs for 5 min. The MagneSil PMPs with bound DNA were robotically transferred by exclusion liquid repellency (ELR) through one PBST (PBS containing 0.1% Tween-20) wash, one PBS wash, and eluted into 15 µL of nuclease-free water (Promega, Cat# P1197). DNA was eluted off of beads for 2 min following manual resuspension. The robot then transferred the MagneSil PMPs out of the elution well, leaving the eluted DNA in water.

**MBD2-MBD enrichment of methylated DNA from CTCs**. The SEEMLIS method of MBD2-MBD enrichment was performed[44]. 25 µL of TALON magnetic beads (Takara, Cat# 635637) were washed 3x with 100 µL 1x Binding Buffer (BB) (4% glycerol, 1 mM MgCl₂, 0.5 mM EDTA, 120 mM NaCl, 2 mM Tris-HCl pH 7.4, 0.2% Tween-20, and 0.5 mM DTT). Beads were resuspended in 100 µL MBD2-MBD Coupling Buffer (1x BB, 1x Halt protease inhibitor cocktail (Thermo Scientific, Cat# PI87786), 500 ng Unmethylated Lambda DNA (Promega, Cat# D1521), and 5 µL tagged MBD2-MBD (EpiXplore Kit, Takara, Cat# 631962). TALON beads and MBD2-MBD are allowed to bind with shaking at RT for 1 h. DNA was digested using 1 µL of each restriction enzyme AluI at 10 units/µL (New England Biolabs, Cat# R0137L) and HpyCH4V at 5 units/µL (New England Biolabs, Cat# R0620L)) in 20 µL reactions containing 1x CutSmart Buffer (New England Biolabs, Cat# B7204S) for 15 min at 37 °C followed by enzyme inactivation for 20 min at 80 °C. MBD2-MBD bound beads were washed 3x with 100 µL 1x BB and resuspended in 88 µL 1x BB with 1x Halt protease inhibitor cocktail and added to 20 µL restriction enzyme digested DNA in 200 µL PCR tubes. This reaction was placed on a shaker at RT for 3 h to bind methylated DNA to MBD2-MBD conjugated TALON beads. PCR tubes were placed onto the Gilson PIPETMAX liquid handling robot (EXTRACTMAN system enabled for ESP as previously described[58]) for washing and elution steps. The robot transferred the whole volume from the PCR tubes onto the extraction microplate (Gilson, Cat# 22100008) and then magnetically transferred the TALON beads through a wash containing 1x BB with 1x Halt protease inhibitor cocktail and into 15 µL of water for elution. The whole elution volume including beads was manually pipetted into new 200 µL PCR tubes containing pre-amplification reaction mix and placed into the thermocycler under manufacturer recommended pre-amplification cycling conditions defined below. Volumes indicated are per reaction.

**Pre-amplification and qPCR of MBD2-MBD enriched DNA from CTCs**. Quantitative PCR was performed using custom TaqMan hydrolysis probes (Applied Biosystems) and iTaq Universal Probes Supermix (Bio-Rad, Cat# 1725153). Primer and probe sequences are listed in Supplementary Data S5. Cycling conditions: 5 min at 95 °C for initial denaturation and enzyme activation followed by 45 amplification cycles of 5 s at 95 °C and 30 s at 60 °C. Pre-amplification was performed using the custom hydrolysis probes and TaqMan PreAmp Master Mix (Applied Biosystems, Cat# 4488593) when indicated according to manufacturer specifications. Cycling conditions: 10 min at 95 °C for enzyme activation followed by 14 cycles of 95 °C for 15 s and 60 °C for 4 min. Pre-amplified samples were diluted 1:5 with TE buffer. Ct values were transformed into relative methylation values by the following equation:

$$\text{Relative methylation} = 2^{-(Ct-MCV)} \qquad (2)$$

Where MCV is the max cycle value, which is the Ct cut off pre-determined by cell line validation studies for each gene. For HLA-I, MCV is equal to 33. For *LINE1*, MCV is equal to 45. A Methylation Index from 0.0 to 1.0 was also calculated for each patient using LNCaP methylation as 1.0 on the Methylation Index scale. Raw Ct values were converted to relative values using the delta Ct method and then divided by the relative methylation in LNCaP cells as follows:

$$\text{Methylation Index} = \frac{2^{-Ct_{CTC}}}{2^{-Ct_{LNCaP}}} \qquad (3)$$

**Statistics and reproducibility**. Survival curve analysis was performed using the log-rank (Mantel-Cox) test. For the HLA-I genomic alteration and HLA-I protein expression studies, comparison between groups was made with an ordinary one-way ANOVA followed by post hoc analysis with the Tukey test for correction of multiple comparisons. Comparisons between average methylation beta values in tumor vs. normal samples from the PRAD data set were made by unpaired *t*-test with Welch's correction. Comparisons between methylation beta values across gene expression subgroups were made with an ordinary one-way ANOVA followed by post hoc analysis with the Tukey test for correction of multiple comparisons. For baseline gene expression and ChIP experiments, comparisons between groups were made with two-way ANOVA using the Dunnett method for correction of multiple comparisons. For drug treatment gene expression experiments, comparisons between DMSO and treatment groups were made by *t*-test corrected for multiple comparisons by the Holm-Sidak method or by one-way ANOVA with Dunnett's correction when comparing DMSO to multiple treatment groups. For CTC MFI experiments, comparisons were made by Kruskal–Wallis test using the Dunn's method for correction of multiple comparisons. Gene expression statistical analyses were performed on delta-Ct values. All statistical analyses were performed in Prism 8 (GraphPad Prism, RRID: SCR_002798). Sample sizes are indicated in the text and figure legends or original documentation in the case of public data sets.

**Reporting summary**. Further information on research design is available in the Nature Research Reporting Summary linked to this article.

## Data availability
Public data sets used in gene expression, methylation, and ATAC-seq analyses are available at cBioPortal (https://www.cbioportal.org), TCGA Wanderer (http://maplab.imppc.org/wanderer/), and UCSC Xena (https://xenabrowser.net). Source data is available in Supplementary Data 8. Uncropped, unedited Western blot images are available in Supplementary Fig. S9.

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

## Acknowledgements

This work was supported by the Office of the Assistant Secretary of Defense for Health Affairs through the Prostate Cancer Research Program [W81XWH-17-1-0096 to T.S.R.] and [W81XWH-12-1-0052 to J.M.L.], the National Cancer Institute of the National Institutes of Health [1R01CA247479-01 to D.J.B. and J.M.L] and a Movember Foundation – Prostate Cancer Foundation Award [17CHAL05 to J.M.L.]. M.C.H. is supported by a grant from the Safeway Foundation. The content is solely the responsibility of the authors and does not necessarily represent the official views of the above mentioned organizations. We would like to thank the patients who donated samples for this study. The results using TCGA-PRAD data sets are based on data generated by the TCGA Research Network: https://www.cancer. gov.tcga. We thank Dr. Bing Yang, clinical research coordinators at UW Urology, and staff at the UWCCC Translational Science Biocore (TSB) for assistance in patient consenting and donor sample acquisition. In addition, we acknowledge NIH grant support for UWCCC Circulating Biomarker Core (CBC), Flow Cytometry and TSB BioBank core services awarded to the University of Wisconsin Carbone Cancer Center (UWCCC) in Support Grant P30 CA014520 and NIH funding for the LSR Fortessa instrument #1S100OD018202-01 and thank all shared resource staff for their valuable contributions to this research. We thank the patients who donated tissue for this study.

## Author contributions

T.S.R. and E.H. wrote and edited the manuscript, designed and conducted experiments, and analyzed data. All other authors reviewed and/or edited the manuscript. C.N.S., C.S.G., K.N.C., M.R.K., A.S., and T.E.G.K. conducted experiments. D.J.B. acquired funding, and designed technology used in experiments. D.F.J. advised on study oversight and design and provided clinical samples. D.G.M. advised on study oversight and design and provided materials for animal studies. M.C.H. designed and conducted experiments, analyzed data, acquired funding, and advised on study oversight and design. J.M.L. advised on study oversight and design, acquired funding, provided clinical samples, wrote and edited manuscript.

## Competing interests

J.M.L. and D.J.B. hold equity in Salus Discovery, which has licensed some of the technology described in the manuscript. D.J.B. also holds equity in BellBrook Labs LLC, Tasso Inc., Stacks to the Future LLC, Lynx Biosciences LLC, Onexio Biosystems LLC, Turba LLC, and is a consultant for Abbott Laboratories. The remaining authors declare no competing interests.
