## [Peer Review File · Communications Biology]

Reviewers' comments:

Reviewer #1 (Remarks to the Author):

In this manuscript, Rodems, Heninger et al. present valuable and timely evidence suggesting that HLA-I expression can be augmented/restore in prostate cancer through epigenetic therapy consisting of DNMT/HDAC inhibitors treatments. However, few elements prevent the publication of this manuscript in its current form or limit the impact of the findings. For example, authors demonstrate that DNMT/HDAC inhibition in LNCaP cells, prior to incubation with PSMA27-38-specific CD8+ T-cells, increases T-cell stimulation and differentiation into cytotoxic T-cells. This finding is not translatable to a possible clinical application since both CD8+ T-cells and tumor cells, as well as the whole organism, would be affected by DNMT/HDAC inhibition. Also, association between HLA-I expression (neg/pos) and the methylation index is inconsistent and there is no robust basis to a possible use of HLA-I methylation index as a biomarker for DNMT/HDAC treatments.

Specific comments:

1) Fig 1: What is the impact of HLA-1 expression on biochemical recurrence in the TCGA_PRAD dataset?

2) Fig 2: In the text, authors mentioned that "This analysis demonstrates a role for DNA methylation in regulating HLA-I expression in primary prostate cancer." However, data provided up to this point is correlative and should be presented as such.

3) Fig 2E: A color key should be provided.

4) Fig 3D/G/H: DNA methylation and H3K27ac/me3 at the proximal enhancer, which was interrogated by ATAC-seq in Fig 2E/F, should be also quantified in Fig 3D/G/H.

5) Fig 3E: Why a lymphoblastoid cell line (GM12878) was used to determine the coordinates of an intronic H3K27ac peak to be probed in prostate cancer cell lines? Multiple prostate-centric H3K27ac/me3 tracks are publicly available and should be used to determine the exact location of the HLA-I peak since regulatory elements can be lineage-specific.

6) Fig 3I: This panel is a "Correlation matrix of HLA-I gene and protein expression, DNA methylation levels, and histone modifications in LNCaP, 22rv1, PC3, LAPC4, and RWPE1". However, LNCaP, 22rv1, PC3, LAPC4, and RWPE1 are not represented in those matrices. Is this a representation of an average value of all cell lines? If this is the case, why RWPE1 are included considering that these are considered as the "normal" prostate epithelial reference? Each cell line should be represented individually or grouped based on specific characteristics.

7) Fig 5: The gene expression data should be complemented with flow cytometry for HLA-I (as performed in Fig 4C) to solidify the link between epigenetics and HLA-I (re)expression.

8) Fig 6: Authors demonstrate that DNMT/HDAC inhibition in LNCaP cells, prior to incubation with PSMA27-38-specific CD8+ T-cells, increases T-cell stimulation and differentiation into cytotoxic T-cells. However, these results are hardly translatable since a clinical approach would require the systemic use of DNMT/HDAC inhibitors (i.e. both the immune system and the tumor compartment, including tumor cells, would be impacted). Therefore, what is the impact of CD8+ T-cell co/pre-treatment with DNMT/HDAC inhibitors on their stimulation and differentiation into cytotoxic T-cells when incubated with LNCaP cells following DNMT/HDAC inhibition? Would this approach work ex vivo or even in vivo?

9) Fig 7G: Association between HLA-I expression (neg/pos) and the methylation index is inconsistent. For example, HLA-A methylation index is lower in HLA-I pos (Pt 568), higher in HLA-I pos (Pt 588; Pt 487) or similar between HLA-I neg/pos (Pt 490). This definitely does not "provides a foundation for future biomarker studies". Authors should make better sense of the data and not over conclude regarding the implications of the presented results.

10) Fig S7: Authors mention that “our ex vivo gene expression analysis suggests a role for APM and B2M epigenetic regulation in prostate cancer patients, which merits further study”. Since B2M is an androgen-regulated secreted protein elevated in serum of patients with advanced prostate cancer (PMID: 17404077), can the author provide B2M serum concentration from patient samples and/or the AR-status and treatment modalities (e.g. ARSI) to contextualize this result?

Reviewer #2 (Remarks to the Author):

The manuscript by Rodems et al showed that DNA methylation and histone modifications are the epigenetic silencing mechanisms behind HLA-1 down-regulation in prostate cancer and that these silencing events can be functionally reversed using DNMT and HDAC inhibitors. Furthermore, the study laid the groundwork for the development of a biomarker based on DNA methylation levels in CTCs to predict epigenetic therapy response in prostate cancer. The study is timely and addresses a very important clinical problem. The methods used in the study are, for the most part, appropriate and clearly described. The major strength of this study is the amount of information which can be used for the development of an epigenetic-based biomarker of prostate cancer immunotherapy response. However, there are a few points that need to be addressed.

1. In the initial investigation of DNA methylation signatures of HLA-1 in prostate cancer using Illumina methylation microarray analysis, it is not clear why the PRAD data set was used. Why was the Taylor data set which contains both primary and metastatic tumor samples not used?

2. Page 5: The authors stated that “... all three genes [referring to HLA-1] tended to have higher methylation in the intragenic regions compared to promoters.” The implication of this result should be discussed.

3. Does the frequency of DNA methylation and histone modification differ between primary and metastatic cancer?

4. Some figures mentioned did not match the corresponding text/statement described:
Page 5: “Correlation patterns varied among the data sets, however we found that DNMT3B was highly negatively correlated ...and HDAC2 was the most negatively correlated to HLA-1 gene expression in the metastatic samples (Figure 1E, S1B).” Figure S1B showed a summary of the genomic alterations in HLA-1 gene and does not support this statement. Do the authors mean Figure 1B?

Page 7: “Since HLA-1 gene expression was associated with a less accessible chromatin state (Figure 2D....” Do the authors mean Figure 2E?

Page 10: “The percent of PSMA ... in each treatment condition is shown in Figure 5C”. Do the authors mean Figure 6C?

5. A few statements in the manuscript in page 4 under the subheading “Aberrantly Expressed DNMT and Class 1 HDAC Genes....” lack proper citation.

Point-by-Point Response to Reviewers

We are grateful to the Reviewers for their in-depth review of our manuscript and insightful suggestions to help improve the presentation and interpretation of our work. We address specific questions below and describe the changes made to the draft and figures to improve the quality of this manuscript.

Reviewers' comments:

Reviewer #1 (Remarks to the Author):

In this manuscript, Rodems, Heninger et al. present valuable and timely evidence suggesting that HLA-I expression can be augmented/restore in prostate cancer through epigenetic therapy consisting of DNMT/HDAC inhibitors treatments. However, few elements prevent the publication of this manuscript in its current form or limit the impact of the findings. For example, authors demonstrate that DNMT/HDAC inhibition in LNCaP cells, prior to incubation with PSMA27-38-specific CD8+ T-cells, increases T-cell stimulation and differentiation into cytotoxic T-cells. This finding is not translatable to a possible clinical application since both CD8+ T-cells and tumor cells, as well as the whole organism, would be affected by DNMT/HDAC inhibition. Also, association between HLA-I expression (neg/pos) and the methylation index is inconsistent and there is no robust basis to a possible use of HLA-I methylation index as a biomarker for DNMT/HDAC treatments.

We thank the Reviewer for the positive comments and appreciate the opportunity to revise this manuscript to address these very important points.

With regards to the co-culture experiments, these studies evaluate antigen-specific T cell recognition of HLA-deficient tumor cells to investigate if epigenetic modifying agents (EMAs) can functionally restore cell surface HLA display and improve T-cell response. This data supports the proposed mechanisms by which EMAs can promote anti-tumor immune re-activation. We agree that translating these concepts into human trials may have significant impacts on immune function that could mitigate the efficacy of re-expression of HLA when combined with checkpoint inhibitors and other immunotherapies, which have had limited success to date in prostate cancer .

Landmark studies have demonstrated that azacitidine (AZA) improves life quality and overall survival in hematologic cancers. The recommended maximum biologically effective dose to treat MDS is 60 mg/m² formulated as a daily sc. injection x 5 with no evidence of significant systemic immune toxicity (NCT01261312). AZA has shown clinical benefit with acceptable toxicity in platinum-resistant ovarian cancer and in metastatic NSLC when combined with entinostat (NCT00387465) (PMID22586682). Although the development of EMA therapeutics have been lagging behind hematologic cancers, there have been concerted efforts at improving bioavailability and reducing toxicity for solid tumor models via mindful re-design of AZA into SGI-110 to enhance the pharmacokinetic profile and ligand-mediated drug delivery of agents

(PMID 32410563) to allow significant dosage reduction. SGI-110 has been reported as a well-tolerated agent in various disease models (NCT01261312, 01696032, 01752933).

Furthermore, epigenetic and transcriptomic profiling has demonstrated that DNA hypomethylating agents boost CD8 T cell activation patterns, cytokine production, cytolytic T cell activity and reverse exhaustion (PMID31548600). In a syngeneic mouse model, HMA treatment led to enhanced CD8-mediated anti-tumor activity, tumor infiltration and suppressed tumor growth. *Ex vivo*, epigenomic and transcriptomic analysis reflected enhanced cytolytic CD8 T cell activity (PMID33609448). In a colon cancer study, SGI-110 plus G-VAX (NCT01966289) enhanced T cell infiltration of the tumors in two patients with no previous baseline (PMID33531075).

A current clinical trial in CRPC and NSLC is assessing anti-tumor activity following SGI-110 in combination with Pembrolizumab (NCT02998567). Anti-tumor responses and epigenetic modification of immune signatures including antigen-expression in the tumor microenvironment are studied in a phase IIb ovarian cancer trial after combination treatment with SGI-110, atezolizumab in the context of CDX-140 vaccination.

In order to fully assess the efficacy of EMAs in improving immune function in PC, a series of future studies are required that include preclinical studies *in vivo* and *ex vivo* to better understand the relevant epigenetic landscape and effects on both tumor and TME elements including immune function. Ultimately, the thorough assessment of EMAs in clinical trials such as the above listed are needed to collect data at a broader range on systemic effects of EMAs beyond the immediate tumor microenvironment.

We have a strong interest in pursuing such studies. We have recently developed a patient-derived-cancer-organoid (PDCO) model system for rapid drug cultures that would allow studying the effect of EMAs in a reconstituted network of patient-derived tumor-microenvironment elements, which would more faithfully reflect individual patient response and the complexity of the tumor niche (Heninger et al. *Med Onc*, 2021, PMID34581895). Furthermore, we also aim to investigate the effect of EMAs in PC in upcoming clinical trials. In ongoing clinical trials, we have been laying the groundwork for this by studying the genomic, transcriptomic and epigenomic landscape of primary, multi-focal prostate cancer (NCT03358563). Clarifying text was added on Page 16.

Regarding the HLA-I methylation we have identified in CTCs, our data provides evidence that this type of analysis is feasible and with further refinement and study, could serve as a clinically relevant approach to monitor DNMT/HDAC treatment effectiveness and/or identify patients who would benefit from such therapies. We agree that HLA methylation in PC CTCs is not an established biomarker ready for clinical study and further research needed to optimize the parameters of the assay for such purpose. Our CTC experiments intended to demonstrate the ability to assess the methylation landscape from rare clinical biospecimens, like liquid biopsy. We maintain that HLA-I methylation could have the potential to help identify patients that would benefit from certain epigenetic therapies, with the assertion that further studies are needed to optimize experimental design, primer locations, patient selection criteria, etc.

Additionally, we have recently established a novel rare-event analysis platform, SEEMLIS (Rodems et al, 2021, accepted for publication at *Clinical Epigenetics*) to capture epigenomic changes in very-low-input samples, such as CTCs. The development of assay is currently in the process of moving towards CLIA-certification at our UWCCC Biomarker Core in order to begin studies to establish clinical translational utility for future use in clinical trials. We have added clarifying text on Page 14.

Specific comments:

1) Fig 1: What is the impact of HLA-1 expression on biochemical recurrence in the TCGA_PRAD dataset?

We thank the Reviewer for this great question. In response to the Reviewers' note, we looked at a similar comparison to Figure 1C using TCGA_PRAD data. In the figure below where Altered Group represents HLA low tumors and Unaltered Group represents tumors that are neither HLA low or high. There is a significant difference in progression free survival for these two groups. This data produces the same message as the Taylor data set, however does not include metastatic tumors since TCGA data consists of only primary tumor samples. We ultimately chose to present the Taylor data set in our figure due to the inclusion of metastatic tumor samples in that data set.

2) Fig 2: In the text, authors mentioned that “This analysis demonstrates a role for DNA methylation in regulating HLA-I expression in primary prostate cancer.” However, data provided up to this point is correlative and should be presented as such.

We appreciate the Reviewer’s notes. We have modified the text accordingly to improve on our interpretations on Page 7: ‘This analysis suggests a role for DNA methylation in regulating HLA-I expression in primary prostate cancer.’

3) Fig 2E: A color key should be provided.

We thank the Reviewer for this suggestion. We replaced the heat map key and updated the figure and figure legend accordingly.

4) Fig 3D/G/H: DNA methylation and H3K27ac/me3 at the proximal enhancer, which was interrogated by ATAC-seq in Fig 2E/F, should be also quantified in Fig 3D/G/H.

We appreciate the opportunity to revisit this figure and provide some updates to improve the accuracy of the figure. Since our original submission, the UCSC Xena browser has updated the available data and the labels associated with the data. We tracked each of the probes (pictured in panels marked D in the images below) using the genomic locations cited by Xena by entering those locations into the UCSC genome browser. Using these genomic locations, we have updated the language in Figure 3 to more appropriately identify the regions that are analyzed by ATAC-seq after reviewing the updated data on the UCSC Xena browser. We updated our figure in the following way: the regions previously labeled as a “proximal enhancer” have been updated to “distal promoter” and the regions previously labeled “promoter” have been updated to “proximal promoter”.

Below are images showing the updated data from the Xena browser, including locations of the ATAC-seq probes. The data below is what is represented in the heat maps in Figure 2E. The ChIP primers that we used in Figure 3 capture the same area as the “proximal promoter” ATAC-seq regions. We have added text on Page 9 ‘This region overlaps with the proximal promoter regions surveyed by ATAC-seq in Figure 2E.’ to reflect this. As an additional note, due to fragment size during chromatin preparation, it is possible that we also capture signal from the distal promoter region as well using the ChIP primers outlined in Figure 3.

5) Fig 3E: Why a lymphoblastoid cell line (GM12878) was used to determine the coordinates of an intronic H3K27ac peak to be probed in prostate cancer cell lines? Multiple prostate-centric H3K27ac/me3 tracks are publicly available and should be used to determine the exact location of the HLA-I peak since regulatory elements can be lineage-specific.

We thank the Reviewer for the opportunity to elaborate on our reasoning for choosing ChIP primer locations. We agree that slight differences can be observed in peak intensity and location

based on cell lineage, however this particular peak is well-conserved among lineage types as can be seen in the image below from the UCSC genome browser. We have updated the text to indicate that this is a conserved peak that is observed by ChIP-seq across multiple cell lineages on Page 9. ‘Primers were designed to locations near a peak in H3K27ac signature that is conserved across multiple cell lineages in GM12878, a lymphoblastoid cell line, determined by ChIP-seq from the ENCODE consortium ³³ (Figure 3E).’ Additionally, because of the proximity of the two H3K27Ac peaks shown here, there is a possibility that we have captured signal from both peaks due to the range of fragment size from shearing during chromatin preparation.

6) Fig 3I: This panel is a “Correlation matrix of HLA-I gene and protein expression, DNA methylation levels, and histone modifications in LNCaP, 22rv1, PC3, LAPC4, and RWPE1”. However, LNCaP, 22rv1, PC3, LAPC4, and RWPE1 are not represented in those matrices. Is this a representation of an average value of all cell lines? If this is the case, why RWPE1 are included considering that these are considered as the “normal” prostate epithelial reference? Each cell line should be represented individually or grouped based on specific characteristics.

We thank the Reviewer for the opportunity to clarify data representation in Fig. 3I. This figure represents a correlation matrix for each of the analyte pairs indicated on the top and left side of the graph. These values represent Pearson correlation “r” values. To clarify, these matrices do not represent averaged cell line data. For example, the value in the bottom left square is the r value that one would generate from graphing protein expression vs. H3K27me3 values in an x vs. y graph. Therefore, each square does include each cell line tested including RWPE1 since we are comparing RWPE1 protein to RWPE1 tri-methylation, LNCaP protein to LNCaP tri-methylation, and so forth to generate the correlation values. By presenting these matrices, we are allowing ourselves and the Reader to see patterns in how each epigenetic factor correlates

with gene and protein expression, as well as how they correlate with each other, in an integrated format instead of looking at multiple individual dot plots.

7) Fig 5: The gene expression data should be complemented with flow cytometry for HLA-I (as performed in Fig 4C) to solidify the link between epigenetics and HLA-I (re)expression.

We are grateful for the Reviewer suggestion. We completely agree on the value of multi-analyte investigation in our *ex vivo* models systems. However, the size of tissue available for rapid drug culture research in the presented patient cohort was limited to ~4mmx4mm core biopsy punches, therefore, the volume of tumor material did not allow for orthogonal analysis of the various treatment conditions in this part of the study, and we were not able to generate matched flow cytometry data for this specific data set. We are currently working on optimizing an alternative approach of rapid drug cultures on human prostate tumor tissue that would allow us to pursue multi-analyte investigation. We have developed a patient-derived prostate organoid model (PMID34581895) that we are able to subject to drug treatment in a microfluidic setting. This platform allows us to generate repeat measurements and investigate multi-analyte read-outs from low-input samples including protein expression, mRNA expression, and DNA methylation patterns utilizing multi-parameter flow cytometry, confocal microscopy, qPCR, transcriptomic and epigenomic analysis. This manuscript is in revisions at Clinical Epigenetics.

8) Fig 6: Authors demonstrate that DNMT/HDAC inhibition in LNCaP cells, prior to incubation with PSMA27-38-specific CD8+ T-cells, increases T-cell stimulation and differentiation into cytotoxic T-cells. However, these results are hardly translatable since a clinical approach would require the systemic use of DNMT/HDAC inhibitors (i.e. both the immune system and the tumor compartment, including tumor cells, would be impacted). Therefore, what is the impact of CD8+ T-cell co/pre-treatment with DNMT/HDAC inhibitors on their stimulation and differentiation into cytotoxic T-cells when incubated with LNCaP cells following DNMT/HDAC inhibition? Would this approach work *ex vivo* or even *in vivo*?

We agree with the Reviewer regarding the limitations in our current preclinical approach to fully understand the effect of EMAs on the complex tumor foci and TME elements including tumor-infiltrating T cells and systemically beyond the TME.

As previous epigenetic and transcriptomic profiling has demonstrated, DNA hypomethylating agents have the potential to enhance tumor immunogenicity, boost effector CD8 T cell responses and other anti-tumor functions including cytolytic CD8 T cell activity and reverse exhaustion (PMID31548600, PMID33609448).

As we discussed above on Pages 1-2, we also have a strong interest in further dissecting these effects preclinically and clinically. We are currently developing novel multicellular *ex vivo* model systems of primary prostate cancer TME that allows the analysis of the effects of EMA-treatment on TME elements beyond tumor cells including immune and stromal components isolated from

clinical tumor biospecimen. The STACKs and LUMENext model systems (PMIDs 33822934, 26610188, 34581895) provide a platform for a reconfigurable multi-cellular network to study complex niches like the TME. The models include an integrated orthogonal platform that enables simultaneous investigations of both cellular interactions and multiple analytes with high-throughput capabilities to assess response to EMAs in protein expression, transcriptomic, epigenomic profiles as well as the secretome. Ultimately, the expansion of focused clinical trial investigations is needed to measure both local and systemic immunological outcomes in PC patients in response to systemic EMA treatment to fully assess the translational value of EMA treatments in restoring MHC I-mediated immune surveillance. We are currently laying the groundwork for this to measure such outcomes in ongoing clinical trials assessing both tumor-infiltrating and systemic immune composition and function in peripheral blood and bone marrow in a cohort of patients with organ-confined prostate cancer undergoing radical prostatectomy (NCT03358563).

9) Fig 7G: Association between HLA-I expression (neg/pos) and the methylation index is inconsistent. For example, HLA-A methylation index is lower in HLA-I pos (Pt 568), higher in HLA-I pos (Pt 588; Pt 487) or similar between HLA-I neg/pos (Pt 490). This definitely does not “provide a foundation for future biomarker studies”. Authors should make better sense of the data and not over conclude regarding the implications of the presented results.

We appreciate the Reviewer’s comments on this experiment and agree that the message is not as clear as we would like it to be to Readers. As we outlined in our earlier response, the goal of this experiment was to demonstrate proof-of-concept that we could analyze HLA methylation in CTCs from prostate cancer patients with the hope of further refining this assay for future clinical use. We certainly do not wish to make the claim that this is a working biomarker or that this is ready for immediate clinical use. Rather, we suggest that this provides a starting point for future studies in the area, demonstrating that this type of analysis is feasible, though refinement and rigorous optimization of analytic methods, primer locations, etc will undoubtedly be required. We do feel that HLA-I methylation has the potential to be useful in the clinic with further refinement, but recognize the limitations in our current results and do not wish to overstep on the implications of our data. We have adjusted the text to better reflect this on Page 14: ‘Overall, this experiment represents the first analysis of HLA-I methylation in prostate cancer CTCs and is a starting point for future studies on methylated DNA biomarkers for prostate cancer therapies.’

Taking into consideration that our mean HLA protein expression level was typically lower than WBC HLA protein expression level, as can be seen in our quantification in Figure 7D, we have adjusted our labeling of CTC populations to “HLA detectable” and “HLA undetectable”. We feel that this is more representative of the data and have updated the figure accordingly, including new representative pictures and updated labels reflecting the new terminology. We have updated the text as well to discuss this in more detail and update the terminology. We feel that this change will make the data more clear for readers and convey the results of our

experiment more efficiently. These changes were made in Figure 7 and supplementary Figure S8 and relevant figure legends.

10) Fig S7: Authors mention that “our ex vivo gene expression analysis suggests a role for APM and B2M epigenetic regulation in prostate cancer patients, which merits further study”. Since B2M is an androgen-regulated secreted protein elevated in serum of patients with advanced prostate cancer (PMID: 17404077), can the author provide B2M serum concentration from patient samples and/or the AR-status and treatment modalities (e.g. ARSI) to contextualize this result?

Excellent suggestion. Unfortunately, matched blood samples are not available for this patient cohort. All the tissue samples included in this study were from patients with organ confined primary prostate cancer without any prior treatment or systemic therapy. This patient group is presenting with localized, hormone naive, AR-positive tumors and elevated PSA. We are planning to do further investigation on a prospective patient cohort with advanced prostate cancer.

Reviewer #2 (Remarks to the Author):

The manuscript by Rodems et al showed that DNA methylation and histone modifications are the epigenetic silencing mechanisms behind HLA-1 down-regulation in prostate cancer and that these silencing events can be functionally reversed using DNMT and HDAC inhibitors. Furthermore, the study laid the groundwork for the development of a biomarker based on DNA methylation levels in CTCs to predict epigenetic therapy response in prostate cancer. The study is timely and addresses a very important clinical problem. The methods used in the study are, for the most part, appropriate and clearly described. The major strength of this study is the amount of information which can be used for the development of an epigenetic-based biomarker of prostate cancer immunotherapy response. However, there are a few points that need to be addressed.

1. In the initial investigation of DNA methylation signatures of HLA-1 in prostate cancer using Illumina methylation microarray analysis, it is not clear why the PRAD data set was used. Why was the Taylor data set which contains both primary and metastatic tumor samples not used?

This is an excellent question and we agree this would be a very interesting analysis. Unfortunately, to our knowledge, methylation array data is not available for the Taylor data set so we are unable to perform this analysis.

2. Page 5: The authors stated that “... all three genes [referring to HLA-1] tended to have higher methylation in the intragenic regions compared to promoters.” The implication of this result should be discussed.

We thank the Reviewer for the opportunity to discuss this further. It has been a paradigm in epigenetics that promoter methylation rather than intragenic methylation traditionally drives epigenetic silencing. While this paradigm is changing with new data supporting important roles for intragenic methylation in regulating expression, we do not yet know the exact implications of this for HLA-I biology. We can speculate that this region is important for loss of HLA-I expression due to the frequency in which we find intragenic methylation among cell lines as well as the correlation of methylation in this region to low gene expression, however without experimentally affecting methylation in the promoter region and/or intragenic region individually, we cannot be sure of the implications. These studies require more complicated techniques such as using Cas9/guide RNA-directed targeting of epigenetic modifying proteins that is beyond the reach of this study. However, this is something we are interested in investigating further in future studies.

3. Does the frequency of DNA methylation and histone modification differ between primary and metastatic cancer?

This is an excellent question. Globally, it has been reported that DNA methylation is lost during prostate cancer progression, while gene specific promoter methylation is increased. This is summarized nicely for methylation in the following paper: PMID: 15026333. Several studies have found that levels of various histone modifications differ between primary and metastatic samples as well. One example for H3K27 methylation in prostate cancer can be found here: PMID: 22149091.

We have added text to the second paragraph of the introduction to give more background on this topic. This can be found on Page 4.

4. Some figures mentioned did not match the corresponding text/statement described:

Page 5: “Correlation patterns varied among the data sets, however we found that DNMT3B was highly negatively correlated ...and HDAC2 was the most negatively correlated to HLA-1 gene expression in the metastatic samples (Figure 1E, S1B).” Figure S1B showed a summary of the genomic alterations in HLA-1 gene and does not support this statement. Do the authors mean Figure 1B?

S1B has been changed to S2B, which shows rank ordered correlation for gene expression and each DNMT and HDAC gene for each study.

Page 7: “Since HLA-1 gene expression was associated with a less accessible chromatin state (Figure 2D....” Do the authors mean Figure 2E?

Thank the Reviewer for the correction, the appropriate reference was Figure 2E. The correction was made accordingly on Page 9.

Page 10: “The percent of PSMA ... in each treatment condition is shown in Figure 5C”. Do the authors mean Figure 6C?

Thank the Reviewer for pointing to this error. 5C has been corrected to 6C, on Page 12.

5. A few statements in the manuscript in page 4 under the subheading “Aberrantly Expressed DNMT and Class 1 HDAC Genes....” lack proper citation.

We have added relevant citations on Page 6 in the first paragraph of the section.

Additional Minor Edits:

1. Bar graphs were updated to showing individual data points in accordance to formatting requirements
2. Minor data corrections were made in Figure 4A, not effecting interpretations.

REVIEWERS' COMMENTS:

Reviewer #1 (Remarks to the Author):

I do not have further comments.

Reviewer #2 (Remarks to the Author):

appropriately revised